# Conservation and divergence of related neuronal lineages in the *Drosophila* central brain

**Ying-Jou Lee, Ching-Po Yang, Rosa L Miyares, Yu-Fen Huang, Yisheng He, Qingzhong Ren, Hui-Min Chen, Takashi Kawase, Masayoshi Ito, Hideo Otsuna, Ken Sugino, Yoshi Aso, Kei Ito, Tzumin Lee***

Janelia Research Campus, Howard Hughes Medical Institute, Ashburn, United States

**Abstract** Wiring a complex brain requires many neurons with intricate cell specificity, generated by a limited number of neural stem cells. *Drosophila* central brain lineages are a predetermined series of neurons, born in a specific order. To understand how lineage identity translates to neuron morphology, we mapped 18 *Drosophila* central brain lineages. While we found large aggregate differences between lineages, we also discovered shared patterns of morphological diversification. Lineage identity plus Notch-mediated sister fate govern primary neuron trajectories, whereas temporal fate diversifies terminal elaborations. Further, morphological neuron types may arise repeatedly, interspersed with other types. Despite the complexity, related lineages produce similar neuron types in comparable temporal patterns. Different stem cells even yield two identical series of dopaminergic neuron types, but with unrelated sister neurons. Together, these phenomena suggest that straightforward rules drive incredible neuronal complexity, and that large changes in morphology can result from relatively simple fating mechanisms.

**\*For correspondence:**
leet@janelia.hhmi.org

**Competing interests:** The authors declare that no competing interests exist.

## Introduction

In order to understand how the genome encodes behavior, we need to study the developmental mechanisms that build and wire complex centers in the brain. The fruit fly is an ideal model system to research these mechanisms. The *Drosophila* field has extensive genetic tools. Furthermore, the relatively small, yet complex fly brain enables scientists to connect neurons into functional circuits, map neural lineages, and test the role of essentially any gene in neurodevelopment and/or behavior (*Venken et al., 2011*). Neurons are wired into neural networks that are connected through neuropils, neurite bundles with dense neurite arborization and numerous synapses. Neuropil structures in adult *Drosophila* brains have been extensively characterized, showing stereotypy and enabling alignment of multiple brains (*Rein et al., 2002*; *Ito et al., 2014*). Further, systematic efforts in cataloging neurons based on morphology and genetic drivers will soon yield a complete neuron type list for the fly CNS (*Aso et al., 2014*). Additionally, the fly brain connectome is being constructed at the level of individual synapses (*Takemura et al., 2017*; *Zheng et al., 2018*). Finally, both morphology and development of the *Drosophila* brain are trackable at the single-cell level. It is therefore possible to resolve fly brain development from neural stem cells to the connectome and ultimately engineer the brain.

The *Drosophila* central brain develops from approximately 100 pairs of bilaterally symmetric neural stem cells, called neuroblasts (NBs) (*Urbach and Technau, 2003*). Most central brain NBs undergo around 100 cell cycles in two neurogenic periods, first building larval networks and then more complex adult neural networks (*Truman and Bate, 1988*). A typical neuronal lineage, born from a single NB, consists of serially derived lineage-specific pairs of post-mitotic neurons. Due to

differential Notch (N) signaling, the paired neurons, made by transient ganglion mother cells (GMCs), can acquire very distinct morphological, physiological, and molecular phenotypes depending on binary sister fates, referred to as A ($N^{on}$) and B ($N^{off}$) fates (*Spana and Doe, 1996*). Within a lineage, neurons of the same A or B fate constitute a hemilineage (*Truman et al., 2010*; *Shepherd et al., 2019*). Temporal patterning allows further neuronal fate diversification within a given hemilineage based on neuronal birth order (*Doe, 2017*; *Miyares and Lee, 2019*). Such lineage-dependent fate specification and diversification underlie the construction of the fly brain connectome.

It is believed that lineages make up individual functional units in *Drosophila*. This was made obvious by marking clonally related neurons produced from individual NBs. NB clones are stereotyped; they have both characteristic cell body positions and distinctive lineage-specific neurite projections. Using MARCM (*Lee and Luo, 1999*) to induce NB clones in newly hatched larvae, we identified ~100 distinct lineages in the fly central brain (*Yu et al., 2013*; *Ito et al., 2013*). Each NB produces lineage-specific neuronal progeny. Discrete lineage identities arise from the spatially patterned neuroectoderm where NB specification occurs, and involve evolutionarily conserved anteroposterior gap genes and dorsoventral patterning genes (*Lichtneckert and Reichert, 2008*; *Urbach and Technau, 2008*). In early embryos, individual NBs can indeed be identified based on combinatorial expression of various transcription factors (TF) (*Urbach and Technau, 2003*). Altering the TF code can transform neuronal fate and wiring, suggesting that lineage identity controls aspects of neuronal type and morphology (*Sen et al., 2014*). Interestingly, fly and beetle have spatially conserved NBs with different TF profiles, further suggesting that NBs and their derived lineages can readily evolve with changes in gene expression (*Biffar and Stollewerk, 2014*).

In order to discern how NBs guide neuronal diversification, we need to appreciate neuronal development at the single-cell level. In other words, we need to map individual neurons back to their developmental origins. Achieving this with stochastic clone induction (i.e. labeling GMC offspring as isolated, single-neuron clones and assigning the neurons to specific lineages based on lineage-characteristic morphology) is possible but laborious and can be extremely challenging for lineages that contain similar neurons. Targeted cell-lineage analysis using lineage-restricted genetic drivers is therefore preferred for mapping specific neuronal lineages of interest with single-cell resolution (*Awasaki et al., 2014*). To date, only three of the about 100 distinct neuronal lineages have been fully mapped at the single-cell level in adult fly brains: the mushroom body (MB), anterodorsal antennal lobe (ALad1) and lateral antennal lobe (ALl1) lineages (*Lee et al., 1999*; *Jefferis et al., 2001*; *Yu et al., 2010*; *Lin et al., 2012*). These mapped lineages consist of 1) MB Kenyon cells (KC), 2) AL projection neurons (PN), and 3) AL/AMMC PNs and AL local interneurons (LN), respectively. All three lineages produce morphologically distinct neuron types in sequential order, indicating a common temporal cell-fating mechanism. However, the progeny's morphological diversity varies greatly from one lineage to the next. The four identical MB lineages are composed of only three major KC types; moreover, paired KCs from common GMCs show no evidence for binary sister fate determination (*Lee et al., 1999*). By contrast, the two AL NBs produce progeny that rapidly change type (producing upwards of 40 neuron types) and the GMCs generate discrete A/B sister fates (*Yu et al., 2010*; *Lin et al., 2012*). In the ALl1 lineage, differential Notch signaling specifies PNs versus LNs (*Lin et al., 2010*). Notably, the paired PN and LN hemilineages show independent temporal-fate changes, as evidenced by windows with unilateral switches in production of distinct PNs or LNs (*Lin et al., 2012*). Moreover, the ALl1 PN hemilineage alternately yields Notch-dispensable AL and Notch-dependent AMMC PNs (*Lin et al., 2012*). Together, these phenomena demonstrate a great versatility in lineage-guided neuronal diversification.

Assembling complex region-specific intricate neural networks for an entire brain requires exquisite cell specificity. In fact, cellular diversity—as characterized by gene expression—is higher during development than in mature brains (*Li et al., 2017*), signifying that the underpinnings of the connectome can be understood by studying development. Such developmental diversity is reflected by characteristic neurite projection and elaboration patterns. We therefore aim to elucidate the roles of NB lineage specification, temporal patterning and binary sister-fate decisions upon neuronal morphology. By doing this, we hope to gain insight about how a limited number of NBs can specify such enormous brain complexity. We chose to map a large subset of NB lineages, enough to make generalizations but not so many to confound analysis. To this end, we selected NBs expressing the conserved spatial patterning gene *vnd* (*Urbach and Technau, 2003*). With single-neuron resolution, we

mapped 25 hemilineages derived from 18 *vnd*-expressing NBs. We observed hemilineage-dependent morphological diversity at two levels. First, neurons of the same hemilineage may uniformly innervate a common neuropil or differentially target distinct neuropils. Second, neurons show additional structural diversity in terminal elaboration, which depends on neuropil targets rather than lineage origins. Once you factor in the differences of the neuropil targets, hemilineages which seem grossly distinct actually show comparable temporal patterns in the diversification of neuron morphology. Many hemilineages exhibit recurrent production of analogous neuron types and/or cyclic appearance of characteristic morphological features, implicating dynamic fating mechanisms. Moreover, we discovered non-sister hemilineages that make similar or even identical neuron types with common temporal patterns. These observations suggest involvement of conserved lineage-intrinsic cell-fating mechanisms in the derivation of diverse neuronal lineages.

## Results

### Mapping 18 neuronal lineages concurrently with *vnd-GAL4*

In order to target a large subset of related neuronal lineages, we wanted to exploit a conserved patterning gene. Both anteroposterior and dorsoventral patterning of the CNS are remarkably conserved from insects to humans (*Lichtneckert and Reichert, 2008*; *Urbach and Technau, 2008*), including the tripartite organization of the brain (the forebrain, midbrain and hindbrain correspond to the fly's protocerebral, deutocerebral, and tritocerebral neuromeres). With the aim of making our findings applicable to all three neuromeres, we searched for a conserved dorsoventral patterning gene with relatively even distribution. Urbach and Technau reported 21 fly brain NBs expressing Ventral nervous system defective (Vnd, homolog of the Nkx2 family of homeobox transcription factors). These include 13 of the 72 protocerebral NBs, 4 of the 21 deutocerebral NBs, and 4 of the 13 tritocerebral NBs (*Urbach and Technau, 2003*). Therefore, we decided to target these Vnd-expressing NB lineages for a detailed, large-scale analysis.

To analyze Vnd$^+$ NBs, we created a GAL4 driver under the control of endogenous *vnd* regulatory sequences using gene targeting (*Chen et al., 2015*). To immortalize the NB expression of Vnd into the neuronal progeny, we derived a Vnd-specific, lineage-restricted LexA driver using a cascade of site-specific recombinases. This cascade is triggered by *vnd-GAL4*, filtered through *dpnEE* (a pan-NB promoter), and then driven ubiquitously so that each of the NB's daughter cells express LexA (*Figure 1A*). To isolate/identify individual Vnd lineages, we utilized stochastic clonal induction of a conditional LexA reporter. We detected 18 stereotyped neuronal lineages with cell bodies within the *Drosophila* central brain (modeled in *Figure 1A* based on data from *Figure 2*). These lineages correspond to the SMPad1, SMPp&v1, SLPpm3, CREa1, CREa2, WEDd1, AOTUv1, AOTUv3, AOTUv4, VLPa2, VESa1, VESa2, ALv1, LALv1, FLAa1, FLAa2, FLAa3, and WEDa1 lineages that we previously identified (*Yu et al., 2013*). The cell body clusters of these *vnd-GAL4* lineages cover the medial part of the anterior brain surface (*Figure 1A* and *Figure 2*), consistent with Vnd's expression around the midline of the embryonic CNS (https://insitu.fruitfly.org/cgi-bin/ex/report.pl?ftype=1&ftext=FBgn0261930).

Despite sharing Vnd expression, the labeled NB clones each show distinct gross morphology. To unravel the extent to which lineage origins and temporal regulation govern neuronal morphology and target selection, we compared the progeny of each Vnd$^+$ NB in detail. We identified individual neurons based on morphology and determined the neuronal birth order for each of the 18 *vnd-GAL4* lineages. We simultaneously mapped all Vnd lineages by twin-spot MARCM (*Yu et al., 2009*) using the Vnd-specific, lineage-restricted LexA driver (*Figure 1*). We conducted transient clone induction in contiguous 2-hr windows from 18 hr after larval hatching (ALH) to 92 hr ALH and from 22 hr before pupa formation (BPF) to 16 hr after pupa formation (APF). We imaged 5771 brains containing twin-spot clones of the 18 Vnd lineages.

Twin-spot MARCM utilizes mitotic recombination to independently label the progeny of a cell division in different colors. This can occur in a cycling NB (labeling the progeny of a GMC in one color and the remainder of the lineage in another) or it can occur in a GMC (labeling the two daughter neurons in different colors). However, not every twin-spot MARCM labeling results in two colors. A clone with only one color can be the result of programmed cell death (PCD). Isolated single-cell clones or single-cell-paired NB clones indicate production of only one viable neuron from a GMC,

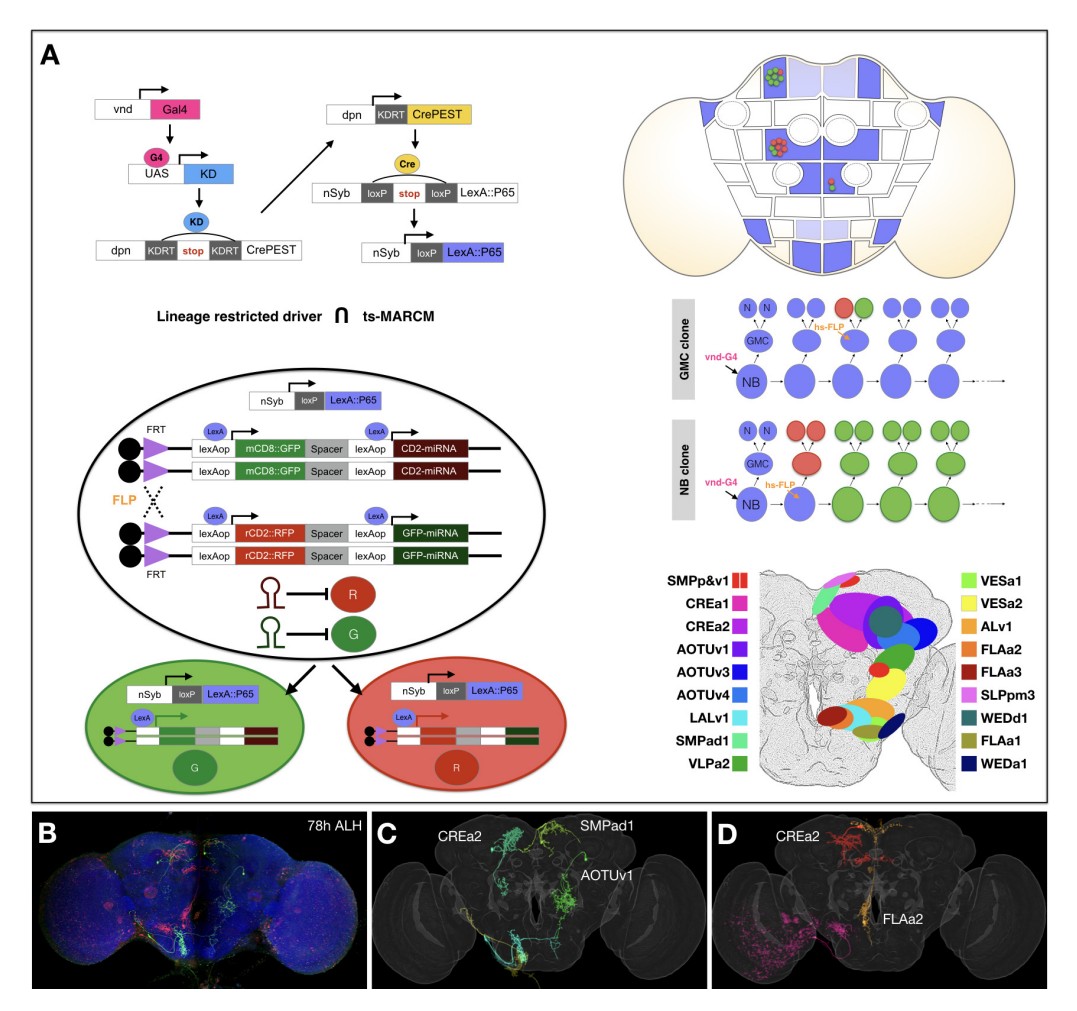

**Figure 1.** Concurrent mapping of Vnd neuronal lineages with twin-spot MARCM. (A) Schematic illustration of twin-spot MARCM with vnd-specific lineage-restricted driver. Top left: LexA driver is restricted to Vnd+ lineages via a multi-step cascade. vnd-GAL4 (pink) reconstitutes dpn-CrePEST (yellow), which in turn reconstitutes nSyp-LexA::P65 (blue). LexA::P65 is only expressed in neurons produced by Vnd$^+$ NBs. Bottom left: mitotic recombination leads to differential labeling of paired sister cells by twin-spot MARCM. Top right: Twin-spot MARCM clones (red/green cells) are stochastically introduced only in LexA::P65 expressing lineages (blue intensity indicates frequency of nSyb-LexA::P65 reconstitution). Middle right: mitotic recombination (hs-FLP) in a GMC elicits paired single-cell clones, whereas recombination in a NB leads to GMC offspring (red) paired with the remainder of the lineage (green). Bottom right: approximate distribution of Vnd neuronal cell bodies in a standard fly brain template. (B–D) Example mapping of multiple twin-spot MARCM clones. A representative nc82-counterstained (blue) adult fly brain (B) carrying multiple twin-spot clones (green/red), induced at 78 hr ALH. The green neurons (C) and red neurons (D) were segmented out and then warped onto a standard adult fly brain. Lineage origin is annotated for neurons with cell bodies in the central brain.

The online version of this article includes the following source data and figure supplement(s) for figure 1:

**Source data 1.** Deconstructing 18 Vnd lineages into serially derived neuron types.
**Source data 2.** Reconstituting full-size NB clones with identified neuron types.
**Figure supplement 1.** Distribution of ~21 k twin-spot clones among 18 Vnd lineages across 59 two-hour windows.

whereas unpaired NB clones indicate PCD of an entire GMC sublineage. We manually annotated individual clones (e.g. *Figure 1B–D*) and ascribed 20,916 clones to specific Vnd lineages in the brain. Although the total clone number for a given lineage varied from hundreds to thousands (*Figure 1— figure supplement 1A*), these differences likely resulted from differential PCD of the progeny or different lineage length. For instance, the recovered VESa1 and VESa2 clones were almost exclusively induced prior to 80 hr ALH, implicating stage-specific progeny production or viability.

In fact, only 7 of the 18 Vnd lineages (SMPp&v1, CREa1, CREa2, AOTUv1, AOTUv3, AOTUv4, and LALv1) produce two viable hemilineages (*Figure 1—figure supplement 1B*). Regardless of

lineage origin, the paired neurons across sister Vnd hemilineages consistently exhibit distinct morphology and have different neuropil targets, reflecting Notch-dependent binary sister fates (*Figure 2A–G*; see below). The remaining 11 lineages exist as a lone hemilineage, as evidenced by the recovery of only unpaired single-neuron clones that share lineage-specific primary trajectories and often innervate common neuropils (*Figure 2H–R*; see below).

To map progeny diversity, we clustered single-neuron clones based on neuron morphology and timing of clone induction. For each of the 18 Vnd lineages, we identified morphologically distinguishable neuron types and further determined their approximate birth sequence based on the recovery window of each neuron type (*Figure 1—source data 1*; *Supplementary file 1*). Single neurons consistently occupied a much more restricted domain compared to full-size NB clones. Moreover, single-neuron clones exhibited birth order-dependent trajectories. Given these phenomena, we examined the extent of lineage coverage in our single-neuron collection by merging representative single neurons from all annotated morphological types and comparing the merged single cells with full-size NB clones. Aligning samples through a standard 3D fly brain template confirmed that all major trajectories from a NB clone were covered by single-neuron projections, ensuring that we have not missed any major neuron types (*Figure 1—source data 2*). Together, our analysis demonstrates that we have systematically mapped the 18 Vnd lineages with single-cell resolution.

## A/B hemilineage-characteristic neurite trajectories

Assigning clonally related neurons to the A ($N^{on}$) or B ($N^{off}$) hemilineage is essential to resolve the impact of binary sister fate decision on neuronal differentiation. Unfortunately, twin-spot MARCM labels paired sister neurons with randomly segregated reporter genes, rather than in a hemilineage-specific manner. Thus, given that Notch underlies hemilineage specification, we genetically manipulated Notch to determine hemilineage identity.

We reduced Notch with RNAi in isolated NB clones (*Figure 2—figure supplement 1A*). Repressing Notch should promote B fate, resulting in reduction of the A-fate morphology and increase in the B-fate morphology. However, when examining NB clones consisting of two sister hemilineages, we noticed that a severe reduction in one hemilineage may not always be accompanied by an evident enhancement of its sister hemilineage. This could be due to death of ectopic neurons because of an incomplete fate transformation or limited space/resources. Further, the loss of neurons with A-fate morphology was incomplete, likely due to weak RNAi. This was particularly obvious among neurons born shortly after clone induction. Regardless of these shortcomings, we could unequivocally determine the Notch state for the 14 paired hemilineages (*Figure 2—figure supplement 1B–H'*). We also made our best possible judgement from available clones on the Notch state of the 11 unpaired hemilineages (data not shown). For consistency, we pseudo-colored neurons derived from A ($N^{on}$) hemilineages in green, and neurons from B ($N^{off}$) hemilineages in magenta when possible (e.g. *Figure 1—source data 1* and *Figure 2*). When referring to hemilineages in text, we added 'A' or 'B' as a suffix (for the seven lineages with two viable hemilineages) or in parentheses (for lone hemilineages).

Knowing hemilineage identity allows us to create hemilineage masks for the seven Vnd lineages producing viable sister-neurons. This was accomplished by clustering single neurons (extracted from twin-spot MARCM clones) based on hemilineage origin. Warping all the individual neurons from the same hemilineage onto a common brain template thus yielded artificial hemilineage clones. Superimposing the hemilineage clones painted in different colors permits close examination of hemilineage distinctions, from cell body distribution, to neurite fasciculation and elaboration (green versus magenta in *Figure 2*).

Notably, the sister hemilineages exist as spatially separate entities in six of the seven paired Vnd lineages. In addition to the SMPp&v1 lineage, in which the hemilineages have noticeably separate cell body locations (see arrows in *Figure 2A*), our analysis reveals hemilineage-specific cell body domains as well as discrete A and B bundles of cell body fibers in the CREa1, CREa2, AOTUv1, AOTUv3, and AOTUv4 lineages (*Figure 2B–F,S–W*). Only the LALv1 lineage has an extensively mixed A/B cell body region and an initial common bundle that extends posteriorly before branching into multiple A- and B-specific fascicles (*Figure 2G and X*).

Regarding terminal elaboration, we see more focused, shorter range innervation in most A hemilineages as compared to their paired B hemilineages. For instance, the LALv1A hemilineage exclusively innervates the central complex (CX). This contrasts the enormous coverage of the LALv1B

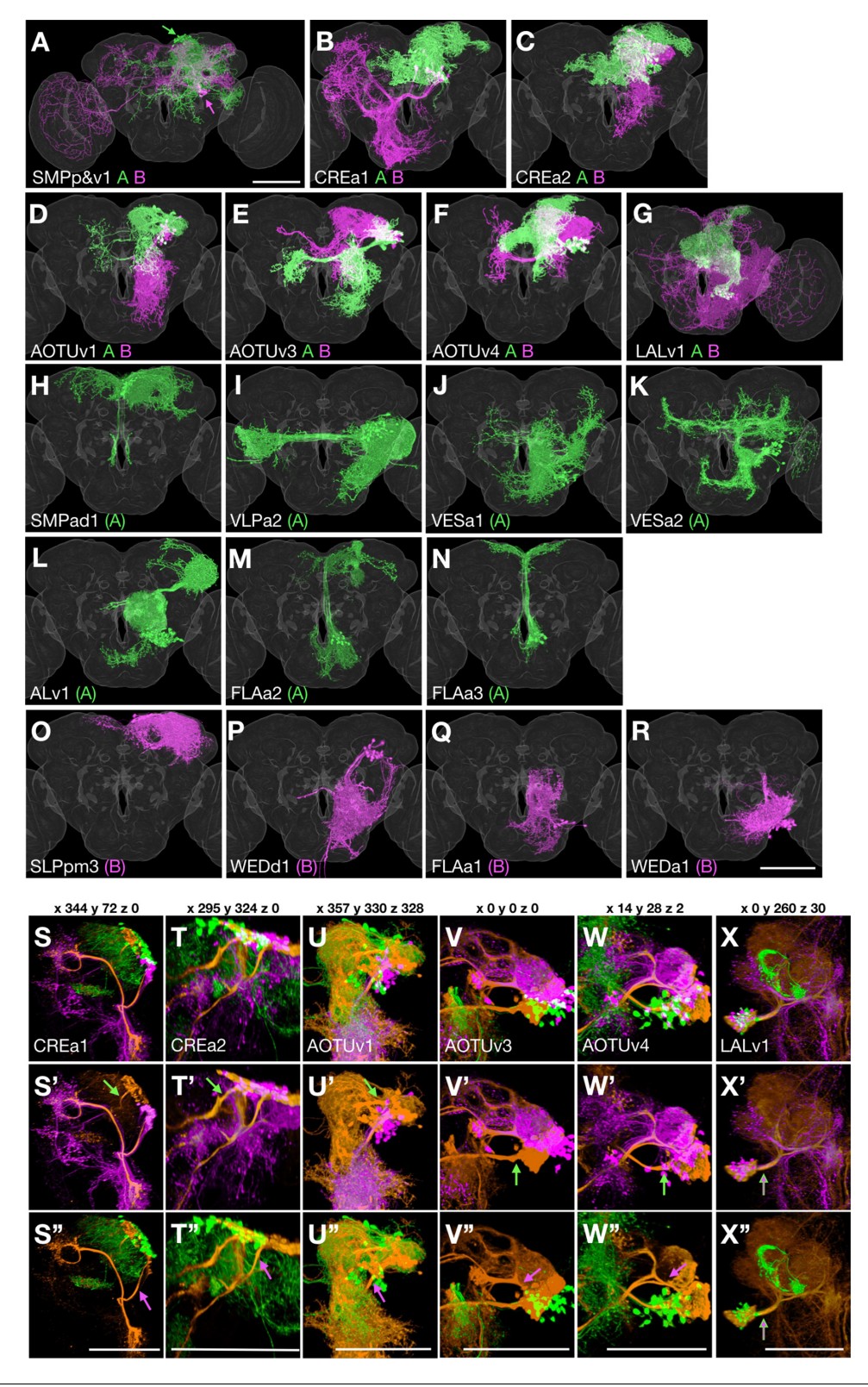

**Figure 2.** Sister hemilineage distinctions. (**A–R**) Hemilineage morphology is revealed by pseudo-coloring $N^{on}$/A neurons green and $N^{off}$/B neurons magenta based on the Notch state (judged from *notch* RNAi phenotypes). For the seven Vnd lineages composed of dual hemilineages (**A–G**), representative single-cell clones of A or B neurons were assembled to create synthetic NB clones with sister hemilineages in distinct colors. For the 11 unpaired Vnd hemilineages (**H–R**), the full pattern was shown by merging the first larval-born neuron with its accompanying NB clone and then pseudo–

*Figure 2 continued on next page*

*Figure 2 continued*

coloring the merged clone according to A/B fate. (**S–X''**) Full-size NB clones (orange) overlaid with both A (green) and B (magenta) hemilineage masks (**S–X**) or either B (**S'–X'**) or A (**S''–X''**) hemilineage mask, to examine the hemilineage-structure correspondence in dual-hemilineage NB clones (except SMPp&v1 with widely separate A/B cell body clusters). Composite confocal images viewed from various angles (x, y, and z coordinates indicated above) demonstrate distinct hemilineage-specific neurite fascicles (indicated with green/magenta arrows), extending out of fully or partially separate A and B cell body clusters, in six of the seven Vnd lineages (**S–W''**). Only the LALv1 lineage has a mixed cell body region that extends a single neurite bundle projecting posteriorly before dividing into multiple fascicles (**X–X''**).

The online version of this article includes the following figure supplement(s) for figure 2:

**Figure supplement 1.** Notch-dependent sister hemilineage projections in seven Vnd lineages.

hemilineage, which spans from the ventral subesophageal zone (SEZ) to the dorsal superior medial protocerebrum (SMP) and even reaches the optic lobe (OL) as well as crossing the midline (*Figure 2G*). The correlation of Notch A/B fate with the extent of neurite targeting and innervation applies to other pairs of A/B hemilineages. For instance, the CREa1A and CREa2A hemilineages elaborate in the vicinity of their cell bodies (green in *Figure 2B and C*). By contrast, their paired CREa1B and CREa2B hemilineages target distant neuropils (magenta in *Figure 2B and C*).

Contrasting the sharp distinctions across paired sister hemilineages, certain non-sister hemilineages look very alike and interestingly share identical Notch states. For instance, the CREa1A and CREa2A hemilineages exhibit grossly indistinguishable morphology (see below). The AOTUv4A and LALv1A hemilineages target same neuropils from distinct cell body clusters. In addition, the B hemilineage of AOTUv1 resembles the unpaired WEDd1(B) hemilineage in both cell body location and in the long-distance ventral targeting (magenta in *Figure 2P and D*).

Given that hemilineages are distinct, our following analyses of neuron diversity independently considered the 25 hemilineages. Nonetheless, in the seven lineages composed of sister hemilineages, we exploited the paired sister-neurons to compare sister-hemilineage development.

## Morphological complexity decreases with birth order

In the process of overlaying single-neuron clones onto full-size NB clones (*Figure 1—source data 2*), we noticed that the first larval-born neurons show uniquely extensive elaborations in five (20%) hemilineages: CREa1B, ALv1(A), VESa1(A), VESa2(A), and FLAa1(B). These first larval-born neurons consistently project farther than the remaining offspring. We thus see extra distant targets exclusively on the GMC side of twin-spot NB clones induced around quiescence exit (*Figure 3*). For instance, the striking LO and SLP innervation by the first larval-born VESa2(A) neuron (green in *Figure 3D*) is completely absent from the largest larval-induced VESa2(A) NB clone (red in *Figure 3D*). In another seven hemilineages (CREa1A, CREa2A, AOTUv1A, AOTUv4A, SLPpm3(B), FLAa2(A), and WEDd1 (B)), multiple early larval-born neurons substantially extend beyond the remainder of the lineage (see *Figure 1—source data 1*). For instance, the postembryonic WEDd1(B) hemilineage consistently starts with two descending neurons that project to the ventral nerve cord (VNC), followed by neurons completely confined to the brain (*Figure 1—source data 1*). All together, we found that 12 out of 25 (48%) Vnd hemilineages contain neurons with uniquely exuberant elaborations born at the beginning of larval neurogenesis.

Contrasting the complexity of early-born neurons, eight hemilineages (32%), SMPad1(A), SLPpm3 (B), CREa1B, CREa2B, AOTUv1B, LALv1B, FLAa2(A), and VESa1(A), terminate with neurons that have drastically reduced domains of elaboration. Four of them, SLPpm3(B), CREa1B, FLAa2(A), and VESa1 (A) (*Figure 4*), produce uniquely exuberant first larval-born neurons, then many neurons with intermediate elaboration, and lastly neurons with limited elaboration. Taken together, our data suggests that the extent of neuronal elaboration is negatively regulated by temporal fate in a similar manner across diverse lineages.

## Neurons of same hemilineage origin vary in topology

We wish to understand to what extent lineage origins determine neuronal morphology and neuronal targets. Considering neuron topology in the context of brain-wide networking, we established a refined neuron classification scheme that takes into account neuron topology and basic topographic features (*Figure 5*). First, we assign the brain-input/output 'extrinsic' neurons, including descending

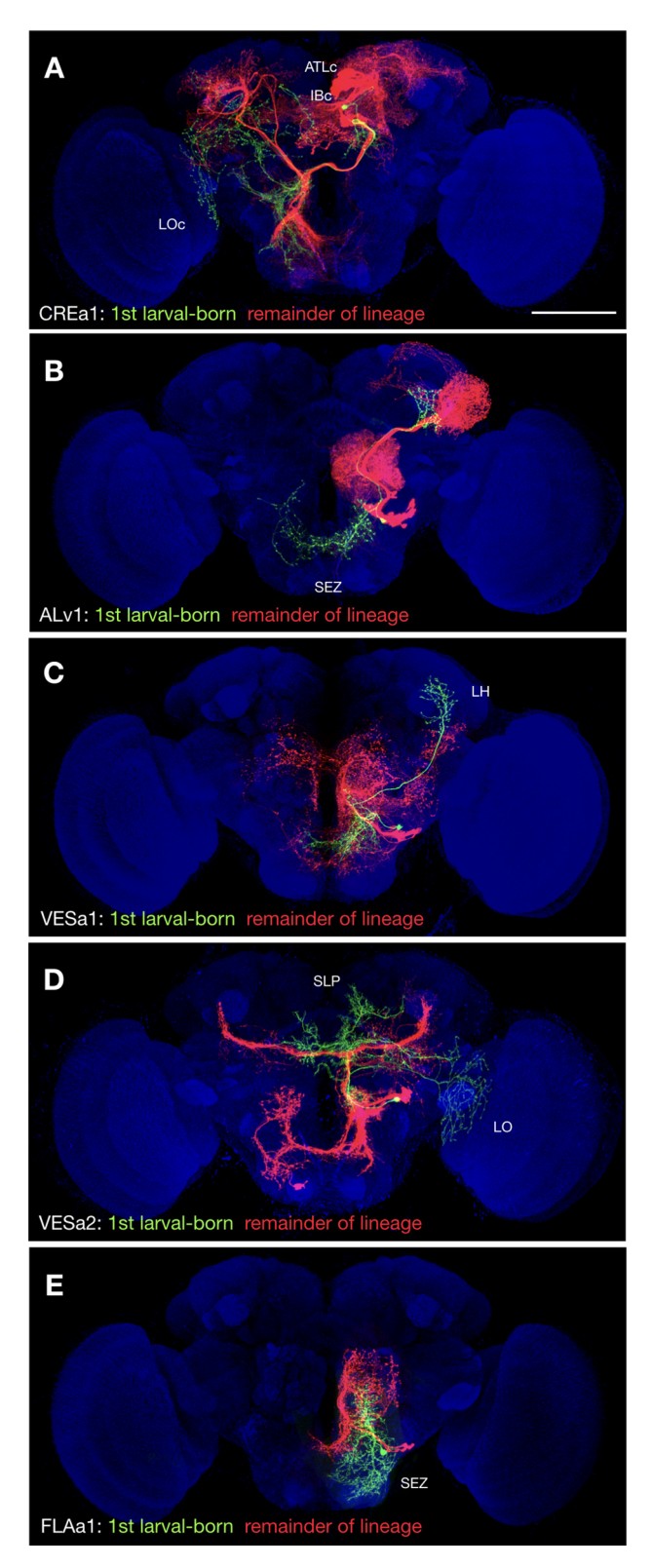

**Figure 3.** Uniquely exuberant first larval-born neurons. (A-E) Representative twin-spot NB clones in nc82-counterstained (blue) adult fly brains. Each twin-spot clone consists of one to two (only CREa1 [A] has two) first larval-born neurons (green) paired with all subsequently born neurons (red) of the same lineage. Neuropils uniquely innervated by the shown first larval-born neurons are indicated.

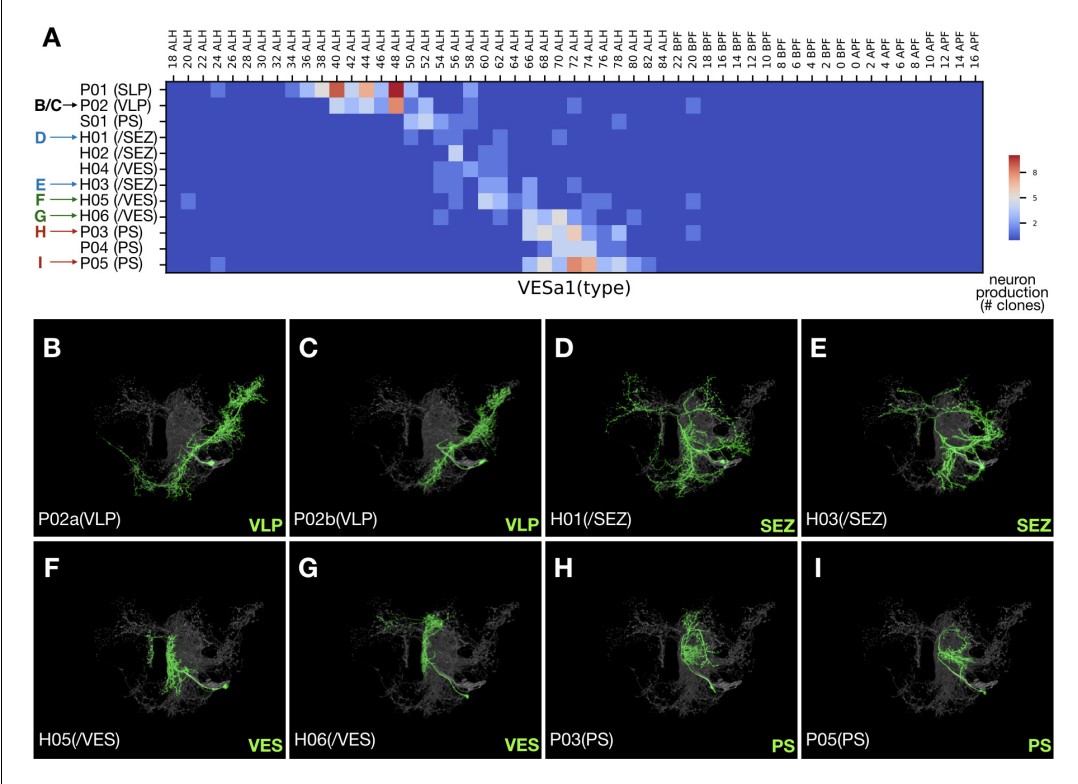

**Figure 4.** Late-born neurons show simplified morphology. (**A**) Heatmap of sample distribution of the annotated VESa1 neurons types (Y-axis) vs. timing of clone induction (X-axis). Blue to red color represents the actual single-cell clone numbers (max = 10) recovered from induction at given time points. Neuron types were manually sorted to reflect their ordered production. Colored arrows indicate distinct morphological groups shown in [B-I]. See text for neuron type nomenclature. (**B–I**) Serially derived single neuron types (green) shown in the context of all recovered VESa1 neuron types (grey). Note reduction in neurite elaboration along birth order.

neurons, to the External cluster. Second, we classify brain-intrinsic neurons based on (1) single or multiple domains of arborization, and (2) unilateral, bilateral, or midline targeting. Briefly, we designate neurons with a single unilateral arborization domain as Single (S), neurons with multiple arborization domains exclusively within one hemisphere as Projection (P), neurons with a single domain of arborization covering the brain midline as Central (C), neurons with midline targeting plus non-midline arborization as Midline (M), and neurons with midline crossing as either Transverse (T) or Horizontal (H) depending on absence or presence of bilaterally symmetric innervation.

We annotated the neurons of Vnd lineages according to this classification scheme and found numerous P-class, H-class, and M-class neurons, several T-class neurons, a few S-class neurons, and only two descending neurons (*Figure 1—source data 1*). Notably, only six (24%) Vnd hemilineages, ALv1(A), CREa2B, FLAa1(B), FLAa3(A), LALv1A, and SMPad1(A), consist of neurons that exclusively belong to a single topological class. If we exclude the uniquely elaborate first-born and austere last-born neurons, we can add three additional hemilineages, AOTUv4A, CREa1B and WEDa1(B), into this list of topologically pure hemilineages. However, there are also six (24%) Vnd hemilineages, AOTUv4B, LALv1B, SMPp&v1A, SMPp&v1B, VESa1(A), and WEDd1(B), composed of three to four topological classes of neurons in addition to the atypical beginning and ending neurons. Taking the SMPp&v1 sister hemilineages as examples, we see P, H and S classes of $N^{on}$ progeny but T, H and M classes of $N^{off}$ progeny (*Figure 6*). Comparing sister hemilineages, the H-class $N^{on}$ and $N^{off}$ cousin neurons display no morphological resemblance despite belonging to the same topological class (e.g. *Figure 6L* versus 6E). By contrast, within a given hemilineage, neurons of various classes often have strong resemblances. For instance, some H-class and P-class SMPp&v1A neurons have nearly indistinguishable elaborations on the ipsilateral side (e.g. *Figure 6O/P and Q/R*). Common ipsilateral morphology also exists among the T, H and M classes of SMPp&v1B neurons (e.g. *Figure 6B–*

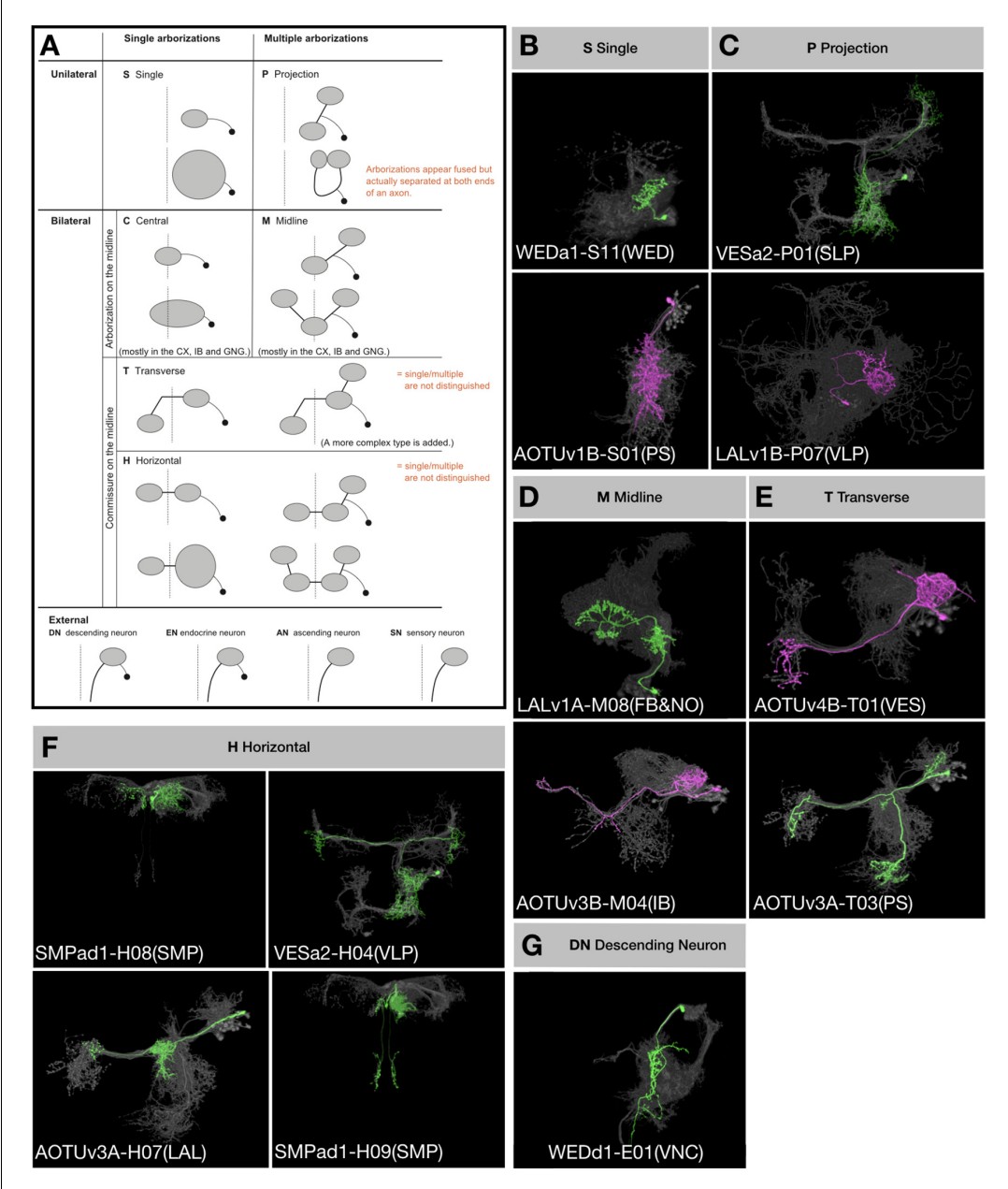

**Figure 5.** Topological classification of single neurons. (**A**) Schematic illustration of neuron morphology classification—multiple examples are given for each class within the brain. Dashed lines indicate the brain midline. Black dots represent cell bodies and grey ovals represent neurite elaborations. (**B–G**) Representative single neurons of each morphological classification (green: A/N$^{on}$, magenta: B/N$^{off}$) shown in the context of all recovered neurons (grey) of the same hemilineages (indicated in the beginning of neuron type names). Multiple neuron types corresponding to each illustrated class are shown, with the exception of Central and External neurons. In the 18 Vnd lineages, no C-topology neurons were found and only the DN type of external neurons was found.

*D*). Moreover, the S-class neurons look like simplified P-class neurons within the SMPp&v1A hemilineage (e.g. *Figure 6I/M*). These phenomena demonstrate the versatility of neurons in adopting various topologies, thus allowing frequent coexistence of H/T/M, P/H, P/S, or H/S classes.

Given the hemilineage-dependent coexistence of selective topological classes, we decided to name clonally related morphological types of neurons with the following convention. Our nomenclature starts with lineage name followed by topology-class letter and a two-digit number. The lineage name is separated from the topology-class letter with a dash sign. To distinguish sister hemilineages,

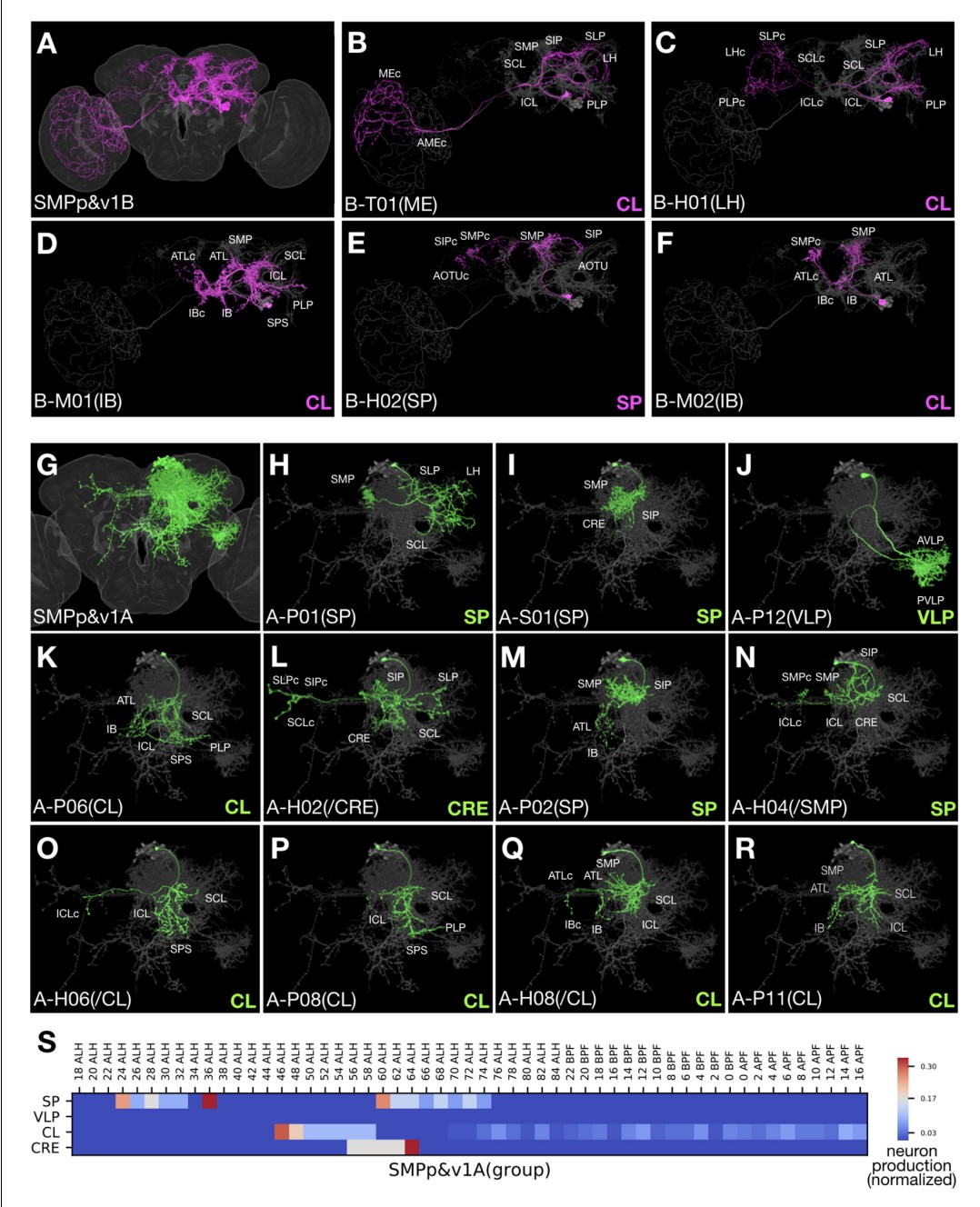

**Figure 6.** Clonally related neurons show diverse topology. (**A**) All identified SMPp&v1B (N^off) neuron types merged onto the standard fly brain template. (**B–F**) Serially derived single neuron types (magenta) shown in the context of all identified SMPp&v1B neuron types merged together (grey). Note presence of T-, M-, H-topolgy neurons with comparable ipsilateral elaborations. (**G**) All identified SMPp&v1A (N^on) neuron types merged onto the standard fly brain template. (**H–R**) Serially derived single neuron types (green) shown in the context of all identified SMPp&v1A neuron types merged together (grey). Note presence of P-, S-, H-topology neurons in the same hemilineage, and similarities among certain S-, P-, and H-topology neurons (**I, M,N**). (**S**) Heatmap of sample distribution of the morphological groups of SMPp&v1A neurons (Y-axis) over time of clone induction (X-axis). The sample distribution was normalized to one for each separate production window. Note presence of two recovery windows for both SP and CL groups, and no detectable recovery window (yielding six or more samples from an uninterrupted interval spanning at least two time points) for the minor VLP group.

we add 'A' or 'B' after the lineage name to represent $N^{on}$ or $N^{off}$. The two-digit serial number starts from 01 and is assigned by arranging neuron types of the same hemilineage-topology class by first clustering neuron types based on closeness in morphology and then determining cluster sequence as well as intra-cluster neuron sequence to roughly reflect birth order. We typically provide extra information in parentheses at the end, to indicate the key morphological feature (e.g. the most distal neuropil target) or simply the preexisting name if available. For H-class neurons with asymmetric neuropil targets, when their key asymmetric target is enclosed as extra information, we add '/' in front of and italicize the enclosed neuropil name. We may add 'i' or 'c' after the neuropil name to indicate 'ipsilateral' or 'contralateral'. Briefly, we have identified 467 morphological neuron types from 25 Vnd hemilineages in the fly central brain (*Supplementary file 2*). The five most heterogeneous hemilineages are: ALv1(A) (46 types), VLPa2(A) (37 types), LALv1B (31 types), LALv1A (29 types), and CREa1A (26 types). The five least heterogeneous are: FLAa1(B) (five types), FLAa3(A) (seven types), CREa1B (nine types), SLPpm3(B) (11 types), and VESa1(A) (12 types). The remaining 15 hemilineages yield 15 to 24 morphological neuron types. As a caveat, we could have over-estimated neuron types due to structural plasticity or we could have under-estimated neuron types due to lack of landmarks, particularly in neuropils that are not well-characterized.

## Hemilineages vary greatly in gross complexity

A 'diverse' hemilineage (one with many assigned neuron types) might consist of distinct neurons that uniformly target the same set of neuropils or grossly dissimilar neurons each innervating distinctive sets of neuropils. Given this phenomenon, we further clustered clonally related neuron types into 'morphological groups' based on patterns of neuropil targeting. For instance, the paired LALv1A and LALv1B hemilineages yield similar numbers (29 vs. 31) of morphologically distinguishable neuron types (*Figure 1—source data 1J*). However, the LALv1A hemilineage produces only one dominant morphological group, whereas LALv1B neurons can be clustered into six different morphological groups (*Figure 7*).

Notably, all LALv1A neurons born after larval hatching project along the same primary track into the CX, including the fan-shaped body (FB), noduli (NO), ellipsoid body (EB), and asymmetric body (AB) (*Figure 7A–Q*). The majority of LALv1A neuron types innervate specific FB layers. These FB neurons are preceded in birth order by several NO- or EB-targeting neurons and followed by multiple AB-targeting neurons. Within the large middle FB window, there are 23 distinguishable neuron types that arise in an invariant sequence (*Figure 7S*). The first three types innervate FB layer 3/4 and ipsilateral NO and show distinguishable proximal elaborations. The fourth neuron type is an outliner, skipping the FB and uniquely targeting the contralateral NO. The next six types selectively innervate a single FB layer, serially targeting layers 2, 4, 7, 8, 6, and 5. The following three types display bilayer targeting, and the next four types selectively target layer 3/4 again. A very similar sequence repeats once, with other multi-layer FB neurons followed by single-layer FB neurons (targeting layers 1 or 2). Interestingly, FB layer 3/4 receives innervation from many more LALv1A neuron types than other FB layers (12 types in total for layer 3/4, as opposed to 2–3 types each for other layers). Regarding the 'proximal' elaborations, multiple rounds of progressive changes in the coverage of LAL/CRE/VES/SP occur along the production of non-FB as well as layer-specific FB neurons (*Figure 1—source data 1J*). One notable example is the recurrent manifestation of the extended dorsal projection connecting SP with specific FB layers (e.g. *Figure 7L and O*).

In contrast to the relatively uniform LALv1A hemilineage, the sister hemilineage, LALv1B, consists of six morphological groups with distinct patterns of neuropil targeting (referred to as IB, PS1, SP, PLP, VLP, and PS2 group based on the principal or unique neuropil of innervation) (*Figure 7A'–R'*). These six major groups arise sequentially (*Figure 7T*). Neurons in a given morphological group can vary in topology, implicating again the versatility in adopting related topologies. However, different morphological groups contain distinct group-characteristic topological classes. This point is reaffirmed by the differential presence of P versus H class in the temporally separated PS1 and PS2 groups. Nonetheless, the PS1 and PS2 groups contain similar-looking S-class neurons, potentially arising through P-to-S versus H-to-S reduction in the PS1 versus PS2 group. Interestingly, the birth of P- or H-class neurons alternates with that of S-class neurons in both PS groups (e.g. *Figure 7C'–F' and O'–R'*). Similar phenomena with recurrent production of comparable neuron types are observed in two additional groups: H-class neurons are born in windows separated by P-class neurons in the PLP group (magenta in *Figure 1—source data 1J11-15*), and vice versa in the VLP group (magenta

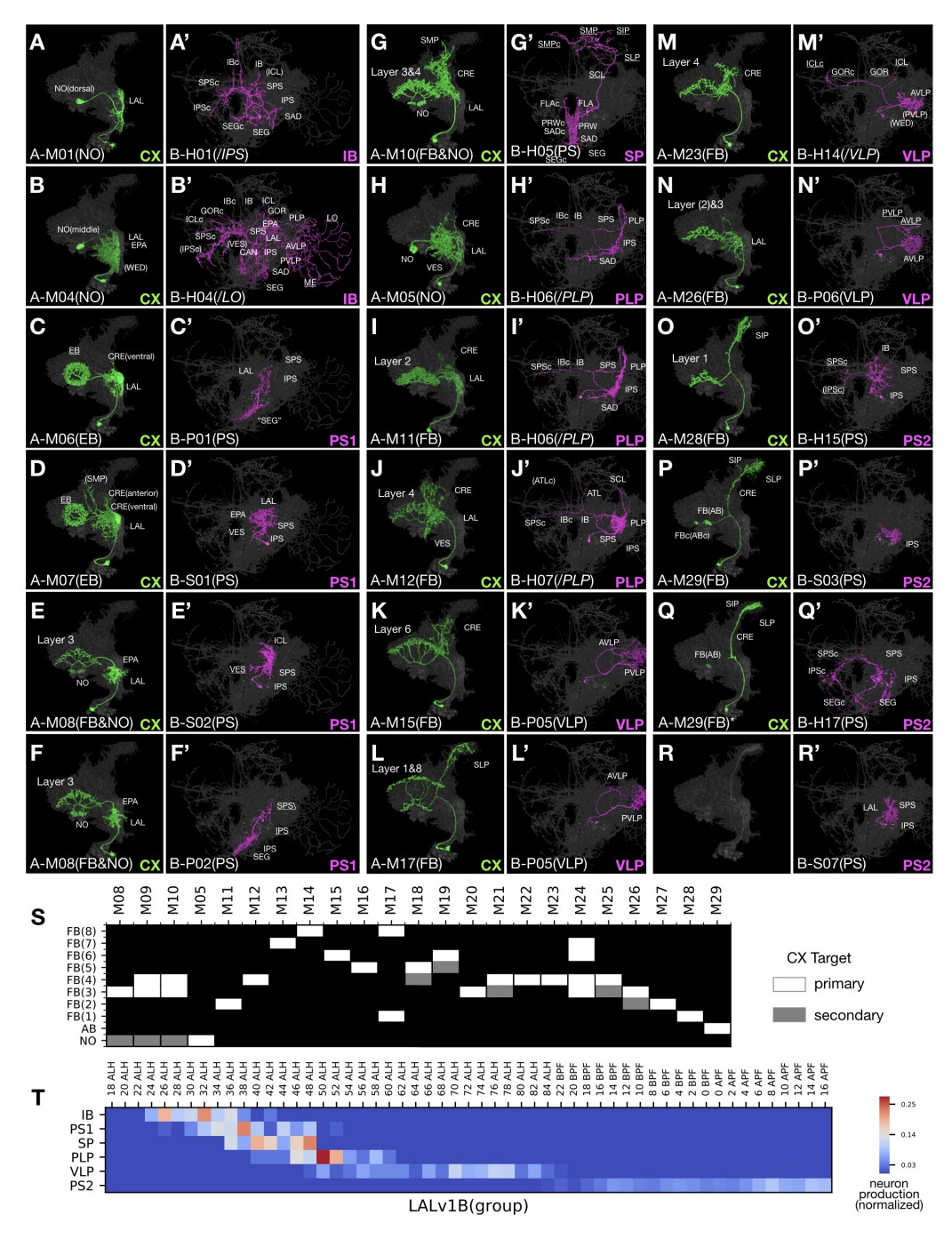

**Figure 7.** Uniform CX neurons pair with diverse groups of non-CX neurons in LALv1. (A–R') Representative pairs of LALv1A (N$^{on}$ green) and LALv1B (N$^{off}$ magenta) neuron types arranged in birth-order, shown in the context of all identified LALv1A neuron types merged together (grey). In the LALv1A hemilineage, there is serial innervation of various CX neuropils in the sequence of NO, EB, FB/NO, NO, then FB (with AB-targeting neurons at the end). Note the recurrent targeting of FB layers ([J] and [M]). The LALv1B hemilineage produces six morphological groups in oder, targeting IB, PS, SP, PLP, VLP, and PS again. Note the presence of P-topology neurons in the PS1 group (C',F') versus H-topology neurons in the PS2 group (O',Q'), as well as similar S-topology neurons in both PS1 and PS2 groups (D',R'). (S) Graphical representation of LALv1A hemilineage CX targets throughput the FB targeting window. Neuron types are arranged in birth-order. (T) Birth-order heatmap of LALv1B morphological groups, demonstrating sequential production of distinct morphological groups.

in *Figure 1—source data 1J16-30*). Such multi-level temporal changes suggest parallel-acting temporal fating mechanisms that run at different paces to diversity neuronal fates in a combinatorial manner.

Moreover, Notch modulates temporal patterning as evidenced by unilateral switches in producing distinct neurons on the A or B side. There exist windows when only one hemilineage is changing types, such that multiple A or B neuron types are paired with a single B or A neuron type (e.g. *Figure 7E–F' and H–I'*). Given the well-defined stereotyped organization of CX sub-compartments, we have high confidence that we identified all individual LALv1A neuron types based on morphology. However, we may have under- or over-estimated the number of LALv1B neuron types. Perhaps, there could exist unidentified types of VLP-targeting LALv1B P-class neurons, as we only distinguished two types of P-class VLP neurons from the same window of time when the sister LALv1A hemilineage contains seven types of FB neurons (*Figure 1—source data 1J16-20 and 29-30*). Conversely, we may have overestimated the number of PS-targeting LALv1B neuron types, especially among late-born neurons which appear more plastic in their morphology (*Figure 1—source data 1J33-45*). Nonetheless, at the level of morphological groups, the paired LALv1A and LALv1B hemilineages show independent temporal patterning despite being derived from the same neural stem cell.

## 12 of 25 hemilineages yield only one dominant morphological group

There are 11 additional (48% in total) Vnd hemilineages which, like LALv1A, contain only one dominant morphological group. Notably, four of the above five most heterogeneous (having the highest number of distinguishable neuron types) Vnd hemilineages, ALv1(A), VLPa2(A), LALv1A, and CREa1A, are among the 12 hemilineages with uniform single-neuron morphology. This paradoxical phenomenon evidently results from the composition of many neuron types involved in constructing fine topographic maps. In particular, the postembryonic VLPa2(A) hemilineage is exclusively dedicated to the formation of the visual topographic map in the VLP neuropil (*Figure 1—source data 1P*). Analogously, all the 46 ALv1(A) neuron types, with the exception of the unique first larval-born neuron, uniformly relay olfactory information from the AL to the LH (*Figure 1—source data 1A*). Besides VLP and LH, the main neuropils targeted by relatively uniform Vnd hemilineages include: the FB (innervated by LALv1A and AOTUv4A), MB lobes (innervated by CREa1A and CREa2A), SP (innervated by SMPad1(A), SLPpm3(B), and FLAa3(A)), PS (innervated by AOTUv1B), WED/SAD (innervated by WEDa1(B)), and VES (innervated by the small FLAa1(B) hemilineage). Interestingly, two thirds of these uniform neuronal series arise from hemilineages with the $N^{on}$ state (see Discussion).

Notably, 10 of the 12 single-group Vnd hemilineages (excluding FALa1(B), and FLAa3(A)) show some recurrent production of similar neuron types. This is most evident in the CREa1 and CREa2A hemilineages which both innervate MB lobes in a repeated and progressive manner, comparable to the cyclic targeting of various FB layers by LALv1A and AOTUv4A (see below). Further, the eight mono-glomerular AL PN types arise sequentially from six discrete windows in the highly heterogeneous ALv1(A) hemilineage (*Figure 8*). As to other lineages, our ability to discern recurrent temporal features could be limited by neuronal targets (with unclear topographic organization) and/or the excessive loss of related neurons.

## 13 of 25 hemilineages yield multiple morphological groups targeting discrete neuropils

There are 12 additional (52% in total) Vnd hemilineages which, like LALv1B, contain multiple neuron groups targeting discrete sets of neuropils. This list includes four unpaired hemilineages (VESa1(A), VESa2(A), FLAa2(A), WEDd1(B)), five paired with a single-group hemilineage (LALv1B, CREa1B, CREa2B, AOTUv1A, AOTUv4B), and two pairs of sister hemilineages (SMPp&v1A/B, AOTUv3A/B). Among them, we see successive neuropil targeting only in four hemilineages and recurrent neuropil targeting (same pattern in multiple windows) in SMPp&v1A, SMPp&v1B, CREa2B, AOTUv3A, AOTUv4B, VESa1(A), VESa2(A), FLAa2(A), and WEDd1(B). Below, we utilize the paired AOTUv3B and AOTUv3A hemilineages to illustrate the successive versus recurrent targeting of discrete neuropils by serially derived neurons.

The AOTUv3B hemilineage yields four morphological neuron groups successively (*Figure 9Q*). The first three groups almost exclusively consist of P-topology neurons, which relay information from

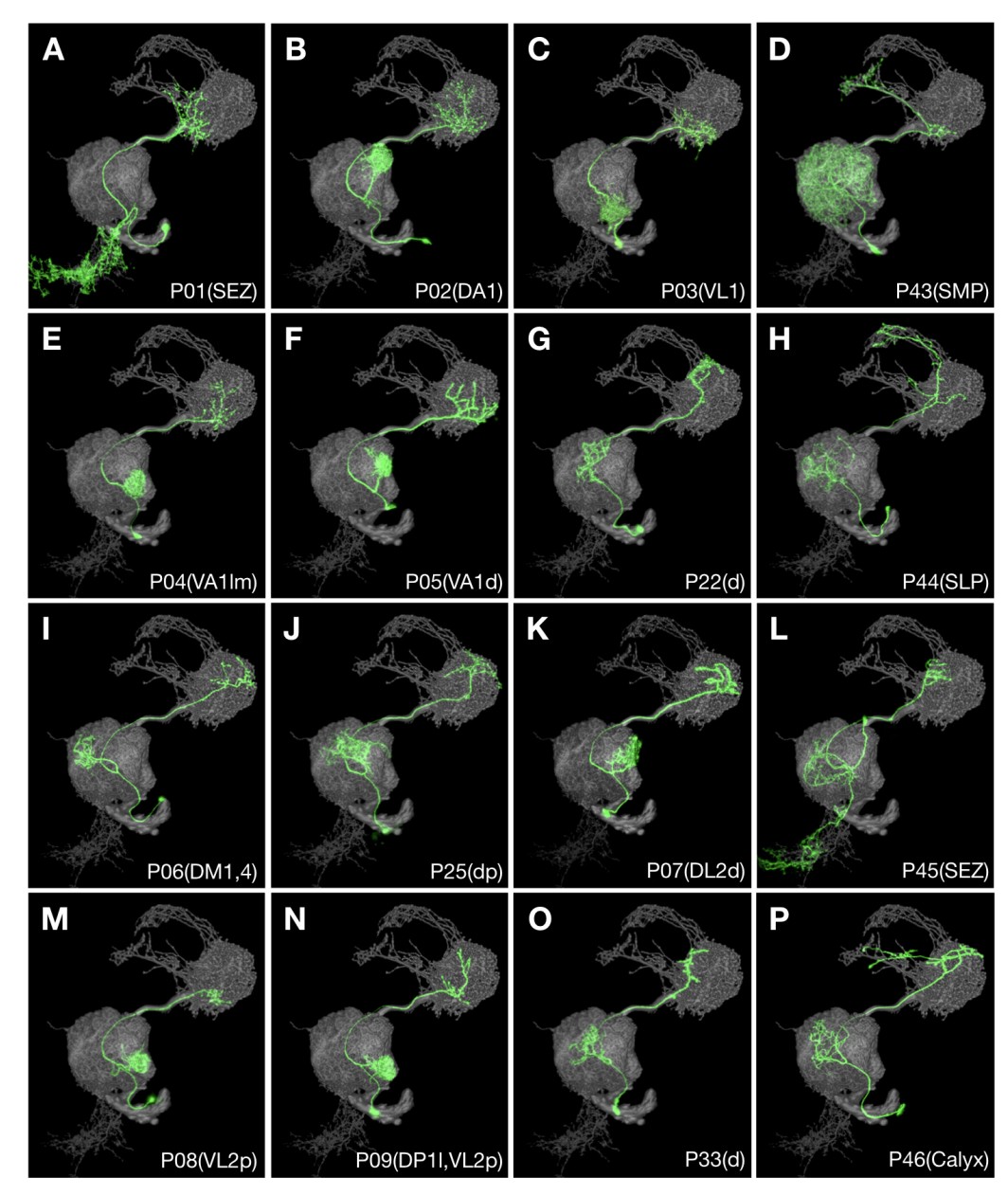

**Figure 8.** Multiple recurrent features in the ALv1 hemilineage. Representative ALv1 neuron types (N^on green) arranged in birth-order, shown in the context of all identified ALv1 neuron types merged together (grey). Note several recurrent features, including mono-glomerular AL innervation (**B, C, E, F, I, K, M and N**), extension beyond LH (**D, H and P**), and SEZ targeting (**A, L**).

the AOTU/SP to the LAL in the first LAL group, from the AOTU to the CRE in the second CRE group, and from the AOTU to the BU in the third BU group (**Figure 9A–F**). By contrast, the fourth IB group only contains M-topology neurons, which connect the AOTU/SP with the SP/IB/PS/ATL (**Figure 9G–H**). However, there lies an 'ectopic' M-topology neuron type born at the transition from the first LAL group to the second CRE group (**Figure 9D**). Nonetheless, this unique M-topology neuron resembles the preceding SP-to-LAL P-topology neurons in other aspects (e.g. **Figure 9C**), reminiscent of versatile neuron topology. Successive changes in morphology further occur within groups. Notably, the proximal elaborations progressively extend from the AOTU to the SP in the first as well as the last group of AOTUv3B neurons (arrows in **Figure 9A–C and G–H**), contrasting the gradual confinement from the SP to the AOTU in the initial group of AOTUv1A neurons (magenta in **Figure 1—**

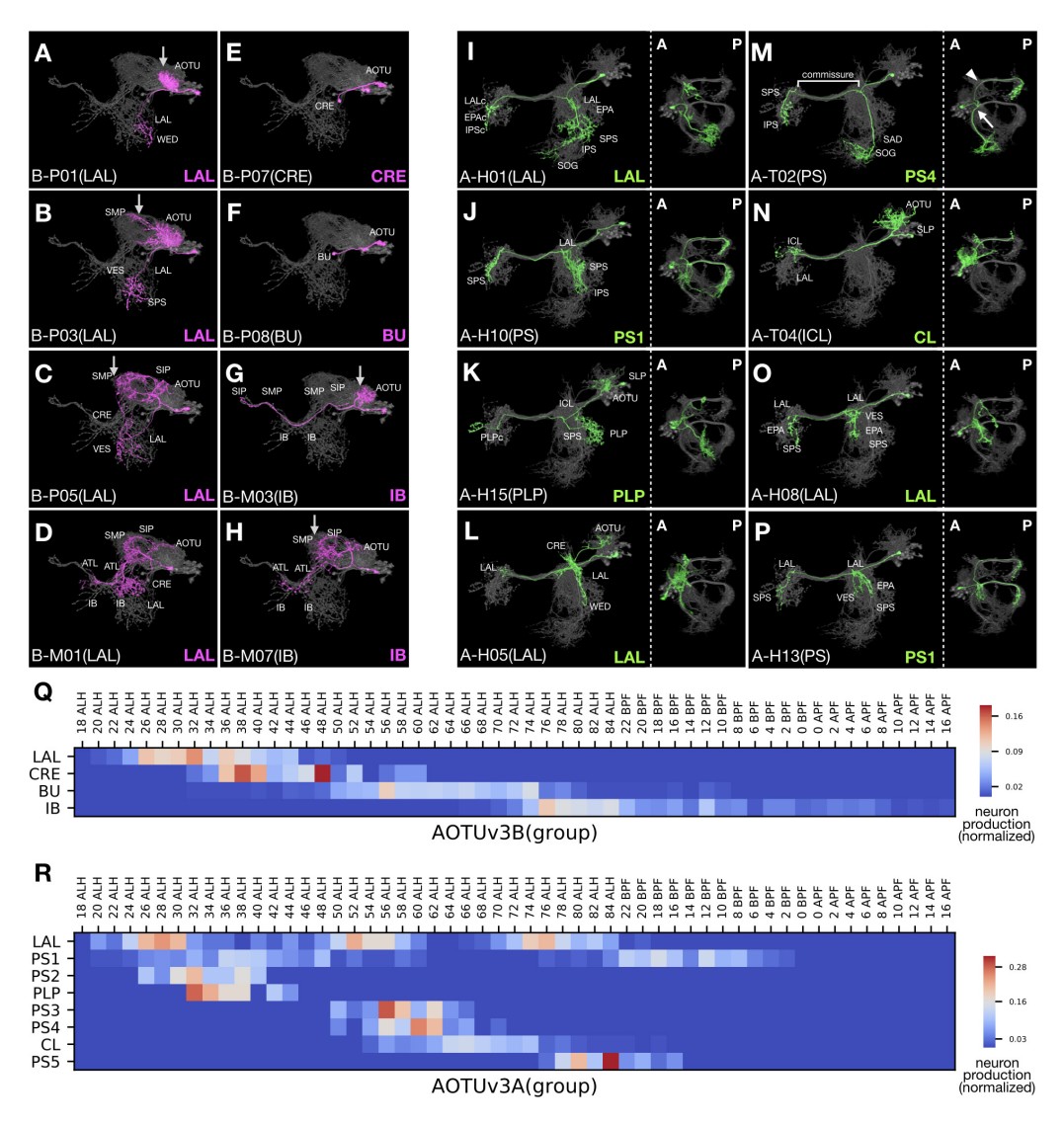

**Figure 9.** Progressive innervation in AOTUv3B and recurrent innervation in AOTUv3A. (A–H) Representative AOTUv3B neuron types (N^off magenta) arranged in birth-order, shown in the context of all identified AOTUv3B neuron types merged together (grey). Note one M-topology neuron type (D) produced much earlier than many other M-topology types (e.g. [G] and [H]). Both LAL and IB groups show progressive AOTU-to-SP proximal elaborations (arrows). (I–P) Representative AOTUv3A neuron types (N^on green) arranged in birth-order, shown in the context of all identified AOTUv3A neuron types merged together (grey). Note multiple recovery windows for LAL (I,L,O) and PS1 (J,P) groups. Tilted views at a lower magnification are showed to the right; anterior (A) to the left, posterior (P) to the right. Note variable length and direction of extensions at the entry and exit of the shared commissure, and recurrence of similar extension patterns (e.g. [J] and [P]). The commissure entry (arrow) and exit (arrowhead) points are indicated in [M]. (Q–R) Birth-order heatmaps of AOTUv3B (Q) and AOTUv3A (R) morphological groups. Note that the LAL group of neurons appear in a single window in the AOTUv3B hemilineage, but multiple windows in the AOTUv3A hemilineage.

source data 1B1-5). In summary, the AOTUv3B hemilineage displays progressive changes in progeny morphology at both intra- and inter-group levels.

By contrast, we see the recurrence of neurons with characteristic morphology in the AOTUv3A hemilineage (*Figure 9I–P*). All AOTUv3A neurons project across the brain midline through the same commissure. Their morphological diversity results from variability in the major branches that extend out at the common entry and exit points of the commissure (e.g. arrow/arrowhead in *Figure 9M*). There are three main configurations of the AOTUv3A neurons' major branches: (1) two posteriorly projecting branches at both entry and exit points (e.g. *Figure 9J and P*), (2) one branch at the exit point (e.g. *Figure 9N*), and (3) no branches (e.g. *Figure 9L*). Various additional features increase

diversity within each configuration, such as ipsilateral AOTU innervation (e.g. *Figure 9L*), ipsilateral or bilateral LAL elaboration (e.g. *Figure 9O*), and a long ventral extension at the entry point (e.g. *Figure 9M*). Notably, there are some T-topology neuron types in this otherwise pure H-topology hemilineage (e.g. *Figure 9M*), which connect the ipsilateral SAD with different contralateral neuropils. As to temporal patterning of neuron morphology, almost all morphological characteristics reoccur multiple times. For instance, we have recovered neurons with bilateral posteriorly projecting branches ('PS1') in three separate temporal windows: at the beginning of larval neurogenesis, shortly after the first, and during the larval-to-pupal transition (e.g. *Figure 9J and P*; *Figure 9R*). In support of long-range temporal patterning (green in *Figure 1—source data 1C*), certain features favor early windows (e.g. AOTU innervation in the first 'PS1' window), some favor late windows (e.g. posterior projection only at the exit point in 'CL' and 'PS5'), and few appear in just one window (e.g. contralateral ICL targeting in 'CL'). In sum, the AOTUv3A hemilineage contains neurons with diverse trajectories in complex temporal patterns.

## Related lineages make similar neurons in comparable temporal patterns

Despite stark differences between sister hemilineages, we see striking similarities between select hemilineages from different NBs. This phenomenon is evident when comparing the sister AOTUv4A and AOTUv4B hemilineages with the unrelated LALv1A and AOTUv3B hemilineages, respectively.

The AOTUv4A hemilineage, like LALv1A, produces only FB-targeting neurons after a short stretch of earlier larval-born non-FB neurons (*Figure 10A–L*). The FB neurons from AOTUv4A and LALv1A are remarkably similar, in both distal and proximal elaborations despite the fact that AOTUv4A targets FB layers 4–8, whereas LALv1A targets all FB layers. We also see similar recurrent targeting of some FB layers in both AOTUv4A and LALv1A hemilineages. Further, the first AOTUv4A FB neuron type exhibits interesting morphological features which are characteristic of the last LALv1A FB neuron type—both show similarly restricted dense elaborations in the SP as well as concentrated FB innervations in small subdomains on either the top (AOTUv4A) or bottom (LALv1A) of the FB (*Figures 10B* and *7Q*). In conclusion, AOTUv4A and LALv1A make similar FB neurons in comparable temporal patterns.

Unlike AOTUv4A, the AOTUv4B hemilineage consists of multiple morphological groups of AOTU-related neurons arising largely in a sequential manner (*Figure 10U*). This is akin to the AOTUv3B hemilineage. The AOTUv4B and AOTUv3B hemilineages have similarities in spatial as well as temporal patterning of neuron morphology. First, both produce P-topology neurons, then many BU-targeting dot-to-dot neurons, followed by midline-crossing neurons (*Figure 10M–T*). Second, in the otherwise pure H- or M-topology group of AOTUv4B or AOTUv3B midline-crossing neurons, there consistently exist a few T-topology neurons (e.g. *Figure 10T*). Third, we see comparable progressive changes in the proximal elaboration from the AOTU to the SP in the non-BU groups of both hemilineages (e.g. arrows in *Figure 10Q–S*). Fourth, in the middle of P-topology neuron production, the appearance of a single neuron type with H- or M-topology occurs in both AOTUv4B and AOTUv3B (*Figures 10O* and *9D*). These extensive parallels between non-sister hemilineages argue for involvement of conserved mechanisms in diversifying neuron fate over time during neurogenesis.

## PAM dopaminergic neurons arise from 'duplicated' lineages

In search of closely related lineages, the neighboring CREa1 and CREa2 lineages have long caught our attention because their full-size NB clones show extensive overlapping elaboration in the MB lobes. However, prior to this study, the identities of neurons innervating the MB lobes were elusive. Briefly, we found that each of the CREa1 and CREa2 NBs produces an intricate sequence of PAM neurons (*Aso et al., 2014*) and that the $N^{on}$ hemlineages, CREa1A and CREa2A, are indistinguishable from each other (compare CREa1A(type) with CREa2A(type) heatmaps in *Figure 1—source data 1*).

Here, we ignore the $N^{off}$ neurons, though CREa1B and CREa2B are always distinctive (magenta in *Figure 1—source data 1E,F*) and therefore instrumental for distinguishing CREa1A from CREa2A. Both CREa1/CREa2 postembryonic NBs start by producing non-PAM $N^{on}$ neurons (green in *Figure 1—source data 1E,F*). The first CREa1A neuron connects the ipsilateral CRE/LAL with the contralateral SMP/ATL/IB (green in *Figure 1—source data 1E*). By contrast, the first CREa2A neuron dies prematurely, leaving its identity unclear. However, the next surviving neuron types in CREa1A

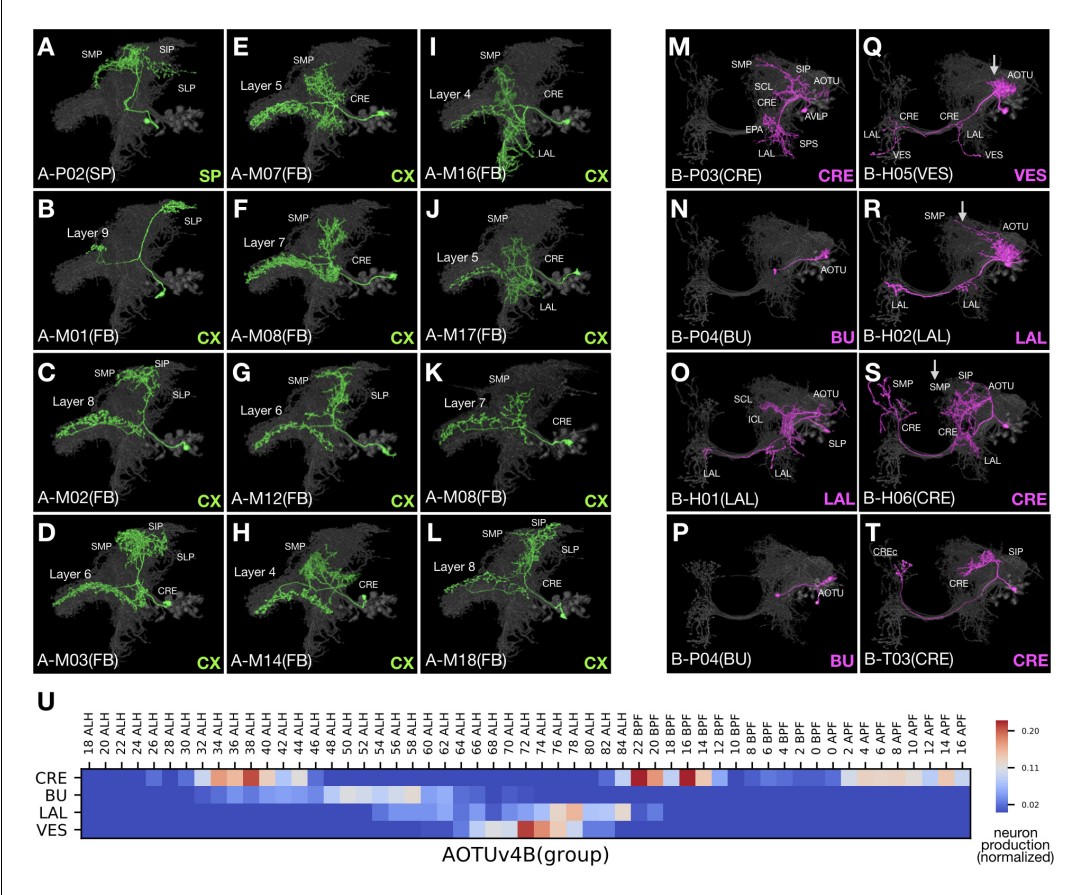

**Figure 10.** AOTUv4A resembles LALv1A and AOTUv4B resembles AOTUv3B. (**A–L**) Representative AOTUv4A neuron types (N^on green) arranged in birth-order, shown in the context of all identified AOTUv4A neuron types merged together (grey). Note the early transition from non-CX neurons (**A**) to FB neurons (**B–L**). Note also that the first FB neuron type innervates layer 9 (**B**), and FB layers 4–8 are recurrently innervated (**C–L**). (**M–T**) Representative AOTUv4B neuron types (N^off magenta) arranged in birth-order, shown in the context of all identified AOTUv4B neuron types merged together (grey). Note one H-topology LAL-group neuron type (**O**) made in the middle of P-topology BU-group types (**N,P**) and much earlier than many other H-topology types (**Q–S**). Both the beginning P-topology CRE-group neurons (see *Figure 1—source data 1D*, magenta in [1] to [4]) and the later multi-group H-topology neurons show progressive AOTU-to-SP proximal elaborations (**Q–S**), as in the LAL and IB groups of AOTUv3B (arrows). (**U**) Birth-order heatmap of AOTUv4B morphological groups.

and CREa2A both innervate the FB (green in *Figure 1—source data 1E6 and 7* and *Figure 1—source data 1F2 and 3*). Strikingly, the CREa1A FB neurons are morphologically indistinguishable from the CREa2A FB neurons. We could tell them apart only by determining their paired sister neurons.

Following the brief production of morphologically identical FB neuron types, both CREa1A and CREa2A NBs yield PAM neurons until they exit the cell cycle. Despite the serial production of different PAM neurons, the CREa1A and CREa2A PAM neurons (identified based on their paired sister neurons) look the same at all time points. This phenomenon indicates that the two NBs produce two identical series of PAM neurons.

In each series of PAM neurons, we identified 17 types of PAM neurons based on MB lobe innervation patterns (*Figure 11A*). This list covers all 14 previously reported types of PAM neurons plus three undocumented morphological types (r4 <r2, r4r5, and b'2r5). In addition, we observe variants of b2, r5, r4, and b'2 p that arise in separate windows and show birth time-dependent patterns of dendrite elaboration (*Figure 11B*). By analyzing birth-order, we see progressive innervation of neighboring zones in the MB medial lobes. a1 is innervated first and the pattern progresses medially along the beta lobe, then laterally (alternating between the gamma and beta' lobes), then medially again along the gamma lobe, and finally ending in b'two2 and b2 (*Figure 11A*). In conclusion, the

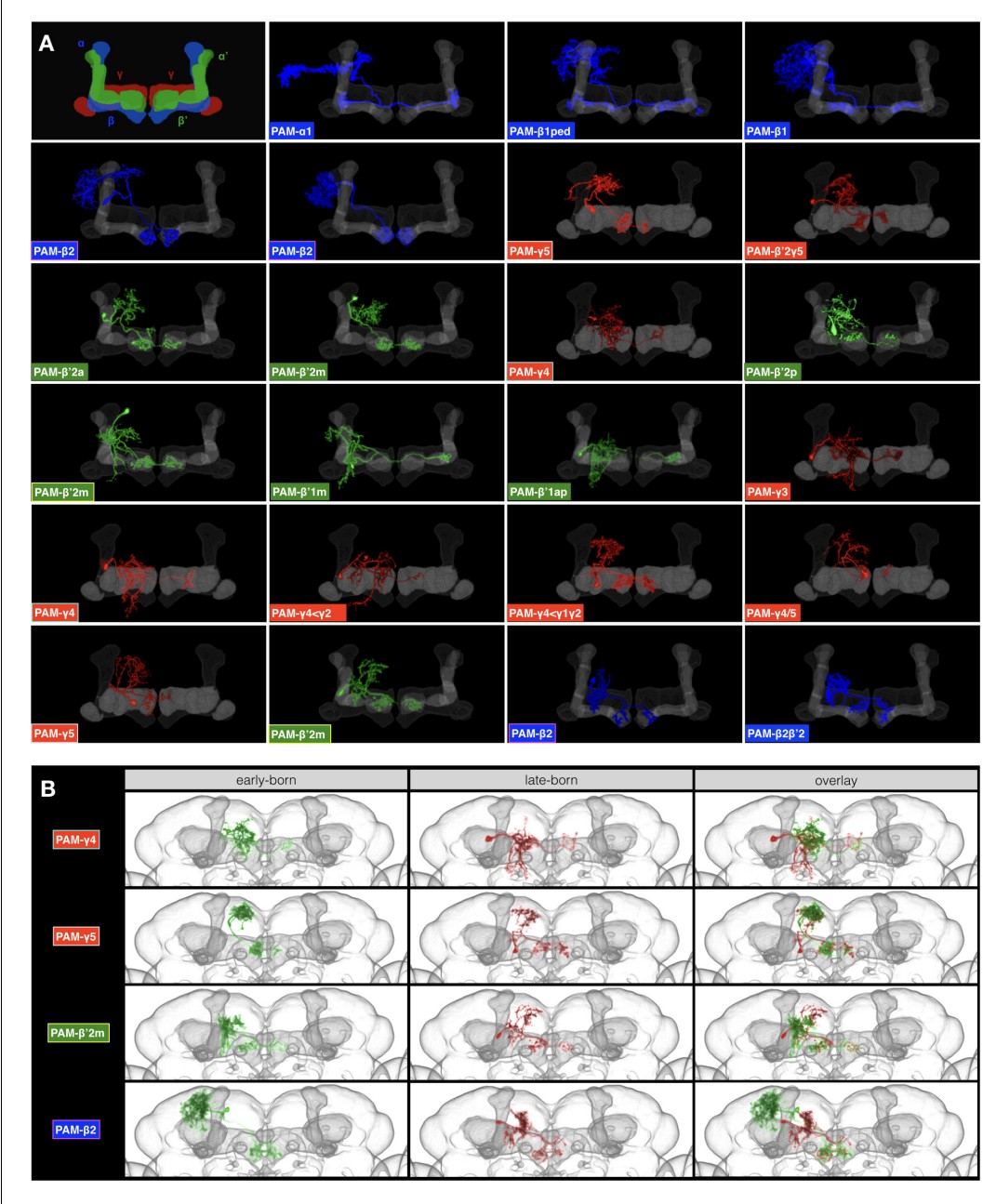

**Figure 11.** Patterned MB innervation by serially derived CREa1A/CREa2A PAM neurons. (**A**) PAM neuronal elaborations (pseudo-colored based on lobe identity) within the MB medial lobes (grey) showing progressive innervation of neighboring MB lobe zones. Note recurrence of four targeting patterns: beta2, gamma5, beta'2m, gamma4. (**B**) Representative PAM neurons (early-born: green and late-born: red) with the same target zone merged onto an adult brain template. Note differential proximal elaborations in early- vs. late-born neurons targeting the same zone of MB medial lobes.

PAM cluster of dopaminergic neurons arises in duplicated series from the non-sister CREa1A and CREa2A hemilineages.

## Single-neuron comparison by NBLAST

Above, we describe the manual annotation of diverse neuronal types based on morphology. This annotation was instructed first by hemilineage identity and then by birth order. In the course of this analysis we could observe patterns that occurred within a hemilineage such as alternate production of neuron types or the recurrence of certain morphological features in complex patterns. With

manual analysis, we could also detect long series of analogous neuron types made by independent hemilineages. However, manual intra/inter-hemilineage comparison is not comprehensive at the single-cell level. We therefore aspired to look for isolated instances of similar neuron types and examine intra-hemilineage morphological diversity in a more systematic manner.

The NBLAST computer algorithm has proven effective in recovering related neurons through pairwise comparison of neuron morphology when preregistered onto a common 3D space (*Costa et al., 2016*). For all-to-all pairwise comparison, we subjected a collection of 464 segmented neurons with one neuron per type (except ALv1_P07, AOTUv4B_H05, and WEDd1_S03 due to poor quality in segmentation) to NBLAST analysis. We selected 326 pairs with similarity scores higher than 0.3 (equivalent to top 3%). To find similar neurons between different hemilineages, we eliminated intra-hemilineage comparisons and recovered 53 pairs of morphologically related non-sibling neuron types. Consolidating interconnected pairs resulted in 16 non-overlapping similarity groups containing 70 Vnd neuron types in total (*Figure 12A*). NBLAST analysis corroborates our manual analysis as eight of the 16 groups can be largely accounted for by the related hemilineages we describe above (CREa1A/CREa2A: 5, AOTUv4A/LALv1A: 2, and AOTUv3B/AOTUv4B: 1). Of the five CREa1A/CREa2A related groups, only two carry an additional neuron not made by the CREa1 or CREa2 NBs. The remaining eight groups were not identified in our manual analysis. In these groups, we see involvement of 16 hemilineages with similarities (for all but one group) limited to a single neuron per hemilineage. Although these cases appear isolated (even after lowering the NBLAST score cutoff to 0.2 and thus doubling the selection to top 7.4%), the grouped neurons do exhibit clear similarities (see *Figure 12B–D* for examples). Seven of the eight isolated cases show extensive overlap in both proximal and distal (groups #1/2/6) or just proximal elaborations (groups #4/5/15/16). However, none of them share common primary trajectory; and a reverse pattern with matched primary projections but distinct terminal elaborations is displayed by the exceptional #14 group. Such incomplete matches reasonably explain our failure in recovering them in our manual analysis. These NBLAST results nicely demonstrate the specificity as well as sensitivity of our manual annotation in detecting closely related neuron types based on overall (rather than local) similarities.

For NBLAST analysis of hemilineage temporal patterning, we clustered the all-to-all pairwise neuronal similarity scores according to hemilineages. When visualizing the NBLAST scores along the order of neuronal birth, diverse temporal patterns of relatedness emerge (e.g. *Figure 12E*). Only two relatively simple hemilineages, CREa1B and FLAa1, display progressive changes in morphology along an entire neuron series. By contrast, various degrees of cyclic changes (indicating recurrent relatedness) exist in all other hemilineages. Patterns of progressive and cyclic changes coexist in six hemilineages: AOTUv4B, CREa1A, CREa2A, SMPp&v1B, SMPad1, and WEDa1. Further, cyclical fluctuations in relatedness (with variable periods) extend throughout the remaining 17 Vnd hemilineages (e.g. CREa2B and SMPp&v1A in *Figure 12E*). Given NBLAST's superb ability to detect local similarities, the widespread phenomenon of cyclic increases in similarity scores indicates the presence of recurrent morphological features. This aligns well with the insights we made above with manual analysis. Taken together, multiple hemilineage-characteristic features may recur at different frequencies to expand the morphological diversity of neurons in a combinatorial manner.

## Discussion

Brain neurogenesis involves conserved patterning mechanisms and employs homologous developmental genes across species (*Huang, 2014*; *Vasconcelos and Castro, 2014*). In *Drosophila*, a series of fating events must occur to promote neuron diversity. First, NBs acquire lineage identity via spatial patterning cues. Second, sequentially born GMCs inherit temporal factors. Finally, GMCs divide into pairs of distinct neurons, generating sister hemilineages with differential Notch signaling. Predetermined fates evidently guide most, if not all, aspects of neuronal differentiation in the invariant fly lineages (*Erclik et al., 2017*).

Despite having predetermined fates, it is hard to imagine how complex neuronal morphology is controlled. Mapping neuron morphology for 25 hemilineages in this study reveals that primary trajectories and thus neuropil targets are mainly dependent upon both lineage identity and Notch signaling—that is hemilineage identity. Notably, sister hemilineages can vary greatly in the extent of innervation. Hemilineages with excessive coverage areas are consistently associated with the $N^{off}$ state (*Figure 2A,B,G*). However, the larger coverage area of the B hemilineage could be a result of

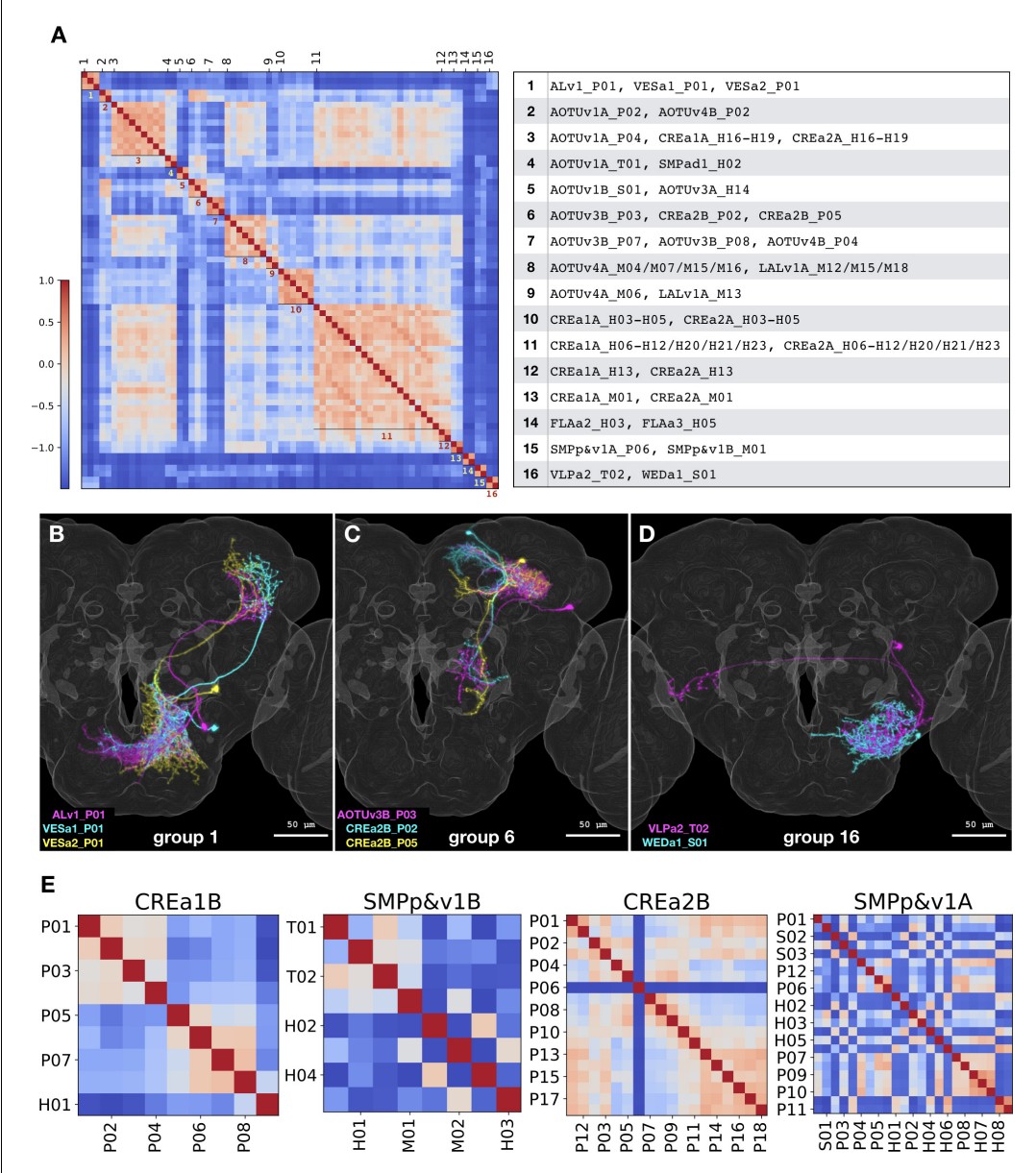

**Figure 12.** NBLAST analysis of inter- vs intra-hemilineage relatedness. (**A**) Sixteen groups of morphologically related neuron types were recovered across independent hemilineages, based on the adjacency matrix derived from NBLAST scores larger than 0.3. Shown side by side are the heatmap of NBLAST scores and the list of Vnd neuron types for the 16 non-overlapping similarity groups. Note that the extensive CREa1A/CREa2A and AOTUv4A/LALv1A inter-hemilineage analogies account for all four (#3/8/10/11) large groups with six or more members. (**B–D**) Representative neurons of the related neuron types for the similarity group #1 (**B**), #6 (**C**), and #16 (**D**). Note extensive overlap in both proximal and distal (B and C) or just proximal neurite elaborations (**D**), despite distinct hemilineage-characteristic primary projections. (**E**) Heatmaps of intra-hemilineage NBLAST scores, sorted based on the birth-order of neuron types (indicated with alternating x-tick and y-tick labels). Note distinct temporal patterns of relatedness in diverse hemilineages. The CREa1B hemilineage displays progressive changes, and the SMPp&v1B hemilineage exhibits both progressive and cyclic changes. By contrast, the CREa2B and SMPp&v1A hemilineages yield more neuron types that appear in various cyclic manners. Please find the complete set of hemilineage NBLAST heatmaps in *Figure 1—source data 1*.

The online version of this article includes the following source data for figure 12:

**Source data 1.** All-to-all NBLAST score matrix of 464 annotated Vnd neuron types.

only a subset of neurons with lengthy projections (e.g. *Figure 13A*). At the single-cell level, the average length of the main trajectory (defined as the total length of the segmented neuron after pruning branches shorter than 50 microns) is significantly greater on the $N^{off}$ than $N^{on}$ side in only two (CREa1 and SMPp&v1, p-value<0.01) of the seven Vnd lineages composed of dual hemilineages (*Figure 13—figure supplement 1*). Instead, we found evidence in support of presence of more diverse morphological groups and/or topological classes of B neurons (as opposed to a dominant group/class of A neurons) (e.g. *Figure 13B*). First, the degree (coefficient) of variation in the length of the main trajectory is significantly higher on the $N^{off}$ than $N^{on}$ side (p-value=0.036 for one-tailed paired T test). Second, there indeed exist significantly higher numbers of topological neuronal classes in the B than A hemilineages (p-value=0.023 for one-tailed paired T test). However, it is unclear if the hemilineage Notch state also affects diversity of neuronal topology in the *Drosophila* thoracic ganglion with well-defined hemilineages (*Shepherd et al., 2019*).

Notch signaling as a binary switch delivers context-dependent outcomes, including grossly opposite phenotypes. For instance, Notch can promote or suppress neuronal cell death depending on lineage identity. This can lead to unpaired hemilineages, in which only one viable neuron is produced after each GMC division. From the 11 Vnd unpaired hemilineages, seven are $N^{on}$ and four are $N^{off}$. Given this random association, it is curious that we see correlation between Notch state and hemilineage complexity. We frequently observe higher gross diversity on the $N^{off}$ side (see above). The same applies to the previously mapped ALl1 lineage where the $N^{on}$ hemilineage consists exclusively

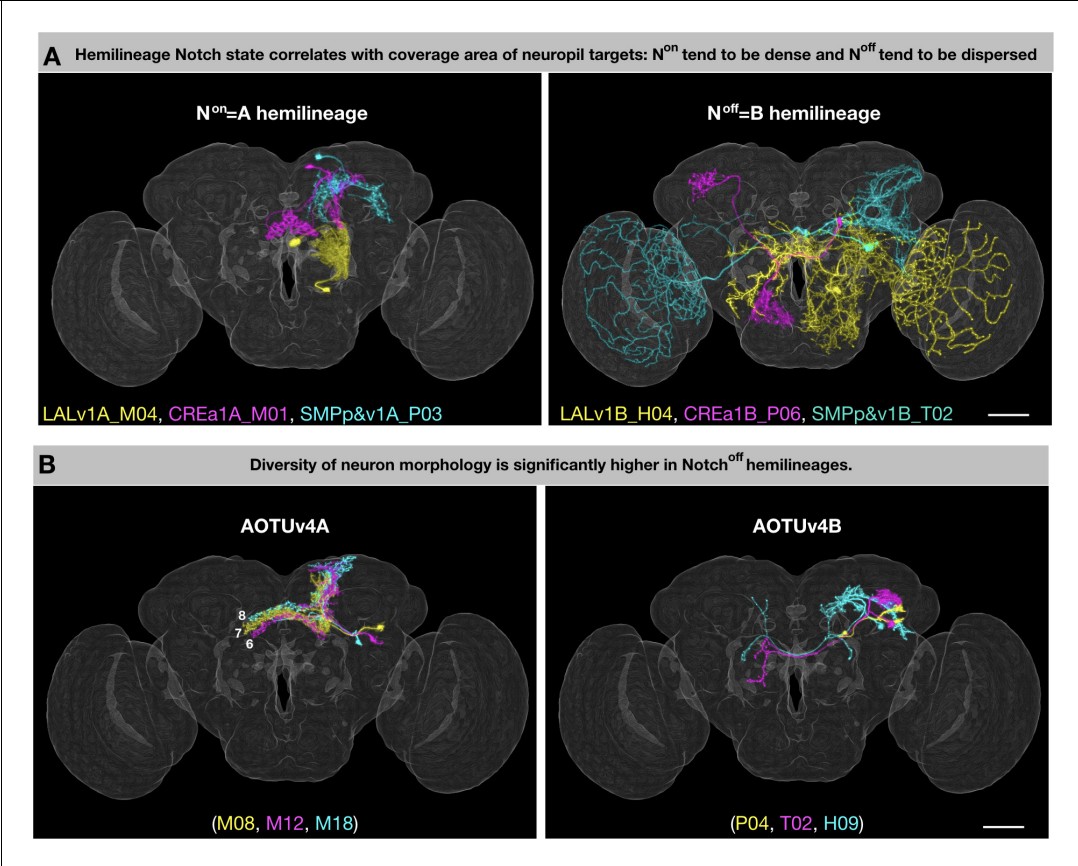

**Figure 13.** General patterns of lineage-guided morphological diversification. (**A**) Notch state correlates with coverage area of neuropil targets. Representative pairs of $N^{on}$/A (i) and $N^{off}$/B (ii) neurons from the LALv1 (yellow), CREa1 (magenta), and SMPp&v1 (cyan) lineages. (**B**) Relatively uniform $N^{on}$/A neurons pair with grossly diverse $N^{off}$/B neurons. Representative pairs of $N^{on}$/A (i) and $N^{off}$/B (ii) neurons, color-coded based on birth order (yellow, magenta, cyan) in the AOTUv4 lineage.

The online version of this article includes the following source data and figure supplement(s) for figure 13:

**Source data 1.** Neuron skeleton lengths of 464 annotated Vnd neuron types.
**Figure supplement 1.** Total length of main trajectory in $N^{on}$/A vs $N^{off}$/B neurons.

of AL local interneurons, whereas its $N^{off}$ sister hemilineage consists of projection neurons innervating diverse neuropils, including AL, AMMC, LH, PLP, and VLP (*Lin et al., 2012*). However, the striking Notch-dependent LN/PN fate separation appears to be a characteristic of only the ALl1 lineage. We found instead that neurons of the same hemilineage can adopt various topologies. In fact, both general topology and terminal arborization seem primarily tailored by the targets innervated (*Figure 1—source data 1*). Further, unrelated hemilineages with striking similarities (e.g. CREa1A/CREa2A and LALv1A/AOTUv4A) consistently have the same Notch state. Such resemblance across non-sister hemilineages could simply reflect their evolutionary relatedness at the lineage level. Nonetheless, Notch may directly regulate neuropil targeting, as implicated by the complete segregation of the $N^{on}$ and $N^{off}$ neuronal processes observed in six of the seven (not LALv1) dual lineages (*Figure 2A–G*). Further, Notch can promote cell adhesion, either by acting as a cell adhesion molecule or by upregulating canonical cell adhesion molecules such as integrins (*Murata and Hayashi, 2016*). Here, we speculate that Notch may strengthen neurite-neurite affinity, as higher affinity in A hemilineages could suppress neurite defasciculation resulting in more uniform trajectory (e.g. *Figure 2B*[green], C[green], F[green], G[green], I, K, L, N), and facilitate extension of long neurite fascicles (e.g. *Figure 2I,K,L,M,N*). By contrast, reduced affinity in B hemilineages could promote gross diversity through serial defasciculation of primary projections. Further, reduced affinity across sister branches could enhance neurite elaboration within targeted neuropils. Nonetheless, additional factors (e.g. neuropil-characteristic topographic maps) might modulate the gross manifestation of Notch's morphogenetic effects.

The orderly derivation of morphologically distinct neuron types within a given hemilineage is indicative of temporal fating. However, the final neuron morphology depends not only on temporal fate, but also on lineage identity and Notch binary sister fate, as well as the anatomy of target neuropils. Despite the complexities, similar temporal features are observable across diverse hemilineages. First, it is common to see beginning neurons with uniquely elaborate projections and ending neurons with reduced morphology (*Figure 14A*). Second, there are temporally ordered neuropil targets characteristic of each hemilineage. Although rarely restricted to a single window, most morphological groups show select windows of production. These phenomena indicate long-range temporal patterning. However, recurrent neuropil targeting is also common. Moreover, a comparable series of related neuron types or progressive morphological changes can appear multiple times in a hemilineage (*Figure 14B*). These recurrences suggest repetition of dynamic factors. Taken together, the temporal changes in neuron morphology and targeting indicate the combination of both long-range temporal patterning and reiteration of temporal windows.

As to underlying molecular mechanisms, the observed birth order-dependent neuronal morphogenesis is unlikely due to the environmental differences over the course of larval neurogenesis, since the final neuronal targeting and innervation occur in a rather synchronized manner at the early pupal stage. Further, some of the temporal patterning phenomena may be explained by the intrinsic temporal factors that have been previously described in the literature. The Cas and Svp embryonic temporal transcription factors are expressed in NBs during early postembryonic neurogenesis (*Maurange et al., 2008*) and are thus candidates to promote the uniquely exuberant neurite projections in first-born neurons. Opposing temporal gradients of Imp and Syp RNA binding proteins in cycling NBs have been shown to control neuronal temporal fate in MB, AL, and complex type II lineages as well as global NB termination (*Liu et al., 2015*; *Ren et al., 2017*; *Yang et al., 2017*). Imp/Syp are likely to govern long-range temporal patterning of most, if not all, neuronal lineages. Imp and Syp gradients shape the descending protein gradient of Chinmo (*Zhu et al., 2006*). The hierarchical temporal gradients of Imp/Syp and Chinmo could define serial temporal windows with expression of various terminal selector genes. For instance, Mamo (a temporally patterned terminal selector gene) is selectively expressed in the window defined by both weak Chinmo and abundant Syp in MB and AL lineages (*Liu et al., 2019*).

However, we do not know exactly how the opposite Imp/Syp protein gradients can define ~30 serial temporal fates in a protracted neuronal lineage. Interestingly, recurrent production of related neuron types has emerged as a dominant theme in the temporal patterning of Vnd lineages. Dynamic Notch signaling may underlie some alternating temporal fates, as Notch has been shown to control alternate production of AL and AMMC projection neurons in the lateral AL lineage (*Lin et al., 2012*). However, it is unlikely that Notch alone can mediate multiple recurring features as seen in most Vnd hemilineages. We therefore propose involvement of parallel recurring factors to

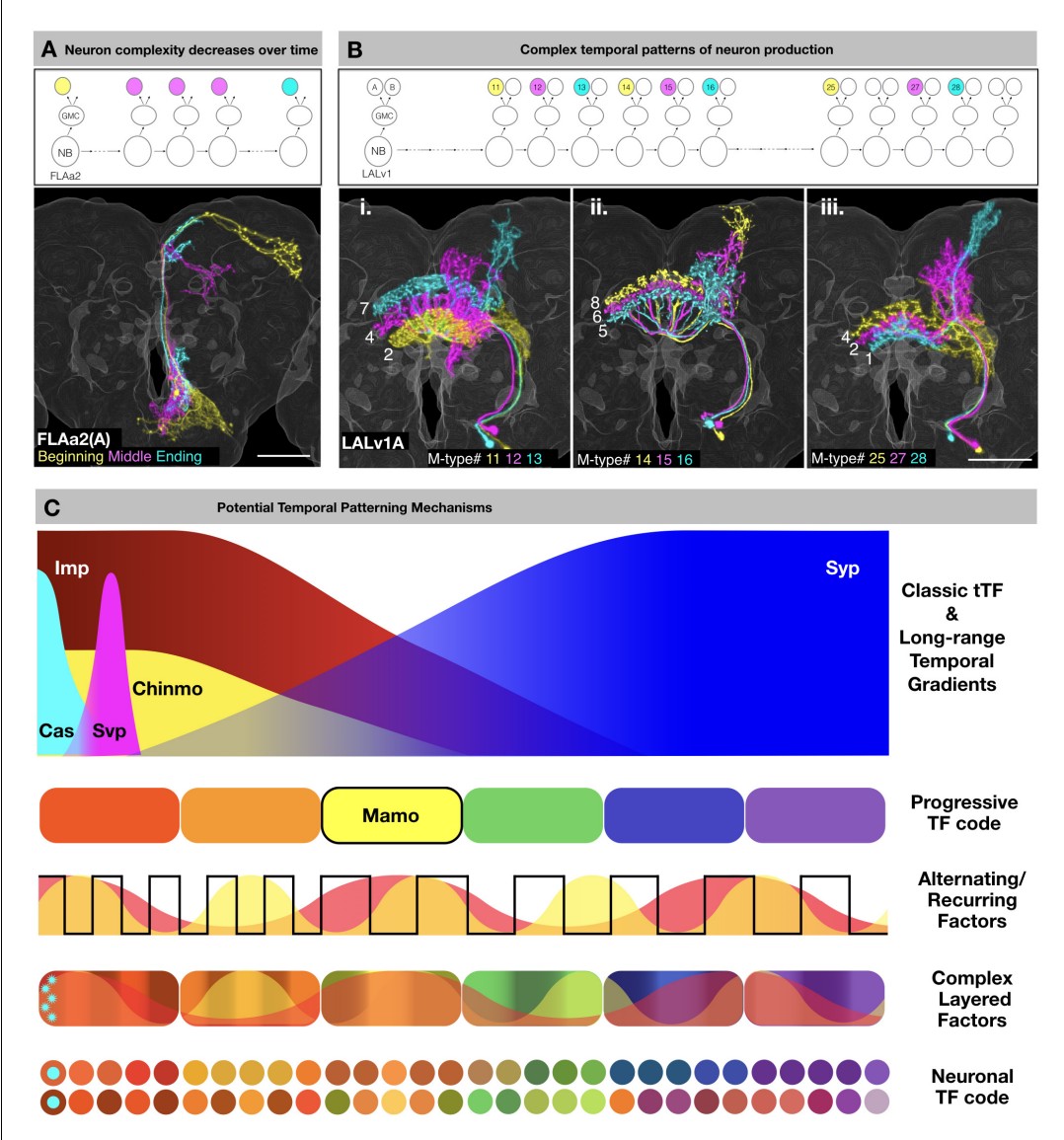

**Figure 14.** Common themes of neuronal lineage temporal patterning. (**A**) Neuron complexity decreases over time. Top: Schematic of FLAa2 postembryonic lineage development showing beginning neuron (yellow), middle neurons (magenta) and ending neuron (cyan). Bottom: Neurons of the FLAa2(A) lineage show gradual restriction in neurite elaboration. (**B**) Complex temporal patterns of neuron production. Top: Schematic of LALv1 lineage development. Select M-type FB innervating neurons are color-coded based on birth order (yellow, magenta, cyan) in three series. Bottom: Serially derived LALv1A neurons with progressive changes in neurite elaboration patterns. Neurons in each panel are color-coded based on birth-order in the same sequence of yellow, magenta and cyan. Note that the ventral-to-dorsal innervation of specific FB layers (i) is followed by reverse dorsal-to-ventral innervation (ii and iii). Also note progressive changes in non-FB elaborations in each round of serial layer-specific targeting. (**C**) Schematic illustration of potential temporal patterning mechanisms, including long-range temporal patterning orchestrated by graded Imp/Syp expressions and asynchronous cyclic changes elicited by unknown alternating/recurring factors. Cas and Svp, as classic temporal transcription factors (tTF), are expressed at the beginning of post-embryonic neurogenesis. Svp triggers the opposite Imp/Syp gradients that in turn shape the descending Chinmo protein gradient. The hierarchical Imp/Syp and Chinmo gradients (plus other undiscovered gradients) could subdivide a protracted neuronal lineage into multiple temporal windows with a progressive TF code (e.g. Mamo). Cas expression (cyan), in the beginning of the lineage may cause the initial exuberant neuron types. In parallel, there could exist various alternating/recurring factors which diversify each temporal window further in a combinatorial manner. Together the tTFs, progressive TF code and alternating/recurring factors lead to a complex layering of factors which are interpreted differentially by sister neurons (depending on Notch state) to reveal the neuronal TF code.

elicit unsynchronized repetition of distinct features. In sum, there likely exist multiple temporal fating mechanisms that act in concert to expand neuron diversity, thus resulting in complex temporal patterns (*Figure 14C*).

Given the need for large neuronal diversity, it was surprising to see the production of two identical, long series of dopaminergic neuron types by CREa1A and CREa2A. Strikingly, the NB homology extends throughout postembryonic neurogenesis. The FB neurons born prior to dopaminergic neurons in CREa1A are also morphologically indistinguishable from those in CREa2A. The only difference is the selective loss of the first larval-born CREa2A neuron. Contrasting the almost identical CREa1A and CREa2A hemilineages, their paired sister hemilineages (CREa1B and CREa2B) are easily distinguishable, as only CREa1B neurons cross the midline. These phenomena implicate that neighboring CREa1 and CREa2 lineages may have arisen from NB duplication followed by a change in midline crossing. Thus, we believe that one way for brain complexity to increase is through lineage duplication and subsequent divergence.

In conclusion, our high-resolution, comprehensive analysis of Vnd lineages reveals how a complex brain can be reliably built from differentially fated neural stem cells. This seminal groundwork lays an essential foundation for unraveling brain development from genome to connectome.

# Materials and methods

**Key resources table**

| Reagent type (species) or resource | Designation | Source or reference | Identifiers | Additional information |
|---|---|---|---|---|
| Antibody | anti-GFP (Rabbit polyclonal) | Thermo Fisher Scientific | Cat # A-11122; RRID:AB_221569 | (1:1000) |
| Antibody | anti-mCD8 (Rat monoclonal) | Thermo Fisher Scientific | Cat # MCD0800; RRID:AB_10392843 | (1:100) |
| Antibody | anti-RFP (Rabbit polyclonal) | Clontech | Cat # 632496 | (1:1000) |
| Antibody | anti-nc82 (Mouse monoclonal) | Developmental Studies Hybridoma Bank | nc82; Registry ID:AB_2314866 | (1:100) |
| Antibody | anti-rabbit, Alexa488 (Goat) | Thermo Fisher Scientific | Cat # A-11034; RRID:AB_2576217 | (1:500) |
| Antibody | anti-Rat, Alexa488 (Goat) | Thermo Fisher Scientific | Cat # A-11006; RRID:AB_2534074 | (1:500) |
| Antibody | anti-rabbit, Alexa568 (Goat) | Thermo Fisher Scientific | Cat # A-11036; RRID:AB_10563566 | (1:500) |
| Antibody | anti-mouse, Alexa647 (Donkey) | Jackson Immuno Research lab, Inc | Cat # 715-605-151 | (1:500) |
| Chemical compound drug | Paraformadehyde 20% Solution, EM Grade | Electron Microscopy Sciences | Cat # 15713 | |
| Chemical compound drug | Phosphate Buffered Saline 10X,Molecular Biology Grade | Thermo Fisher Scientific | Cat # 46–013 CM | |
| Chemical compound drug | Triton X-100 | Sigma-Aldrich | Cat # 329830772 | |
| Chemical compound drug | SlowFadeTM Gold antifade Mountant | Thermo Fisher Scientific | Cat # S36936 | |
| Chemical compound drug | Ethyl alcohol, pure | Sigma-Aldrich | Cat # 459844 | |
| Chemical compound drug | Xylenes | Thermo Fisher Scientific | Cat # X5-500 | |
| Chemical compound drug | DPX mountant | Electron Microscopy Sciences | Cat # 13512 | |

*Continued on next page*

*Continued*

| Reagent type (species) or resource | Designation | Source or reference | Identifiers | Additional information |
|---|---|---|---|---|
| Genetic reagent (*D. melanogaster*) | UAS-Notch-RNAi | Bloomington *Drosophila*stock center | BDSC:33611; FBti0140084; RRID:BDSC_33611 | FlyBase symbol:P {TRiP.HMS00001}attP2 |
| Genetic reagent (*D. melanogaster*) | UAS-mCD8-GFP | *Lee and Luo (1999)* | | |
| Genetic reagent (*D. melanogaster*) | hs-ATG > KOT > FLP | *Ren et al., 2018* | | |
| Genetic reagent (*D. melanogaster*) | dpn > FRT-STOP-FRT>Cre::PEST | *Awasaki et al. (2014)* | | |
| Genetic reagent (*D. melanogaster*) | lexAop2-rCD2::RFP-insulator-lexAop2-GFP-RNAi, FRT40A | *Awasaki et al. (2014)* | | |
| Genetic reagent (*D. melanogaster*) | hs-FLP,dpn > KDRT-stop-KDRT>Cre PEST; lexAop2-mCD8::GFP-insulator-lexAop2-rCD2-RNAi, FRT40A; nSyb > loxP-stop-loxP>LexA::P65,UAS-KD1 | *Awasaki et al. (2014)* | | |
| Genetic reagent (*D. melanogaster*) | vnd-T2A-Gal4 | This paper: Materials and methods | | Lee T, Janelina Research Campus, HHMI |
| Genetic reagent (*D. melanogaster*) | ase-KD1 | This paper: Materials and methods | | Lee T, Janelina Research Campus, HHMI |
| Genetic reagent (*D. melanogaster*) | act > loxP-STOP-loxP>Gal4 | This paper: Materials and methods | | Lee T, Janelina Research Campus, HHMI |
| Software and Algorithms | Fiji | NIH; *Schindelin et al., 2012* | https://fiji.sc | |
| Software and Algorithms | Adobe Photoshop | Adobe Systems, San Jose, CA | https://www.adobe.com | |
| Software and Algorithms | Adobe Illustrator | Adobe Systems, San Jose, CA | https://www.adobe.com | |
| Software and Algorithms | Python | Python Software Foundation | https://www.python.org | |
| Software and Algorithms | Flybase 2.0 | *FlyBase Consortium et al., 2019* | http://flybase.org | |

## Fly strains and DNA constructs

Transgenes used for twin-spot MARCM for vnd lineages include: vnd-T2A-GAL4 (this study), UAS-KD, dpn >KDRT-stop-KDRT>Cre:PEST, nSyb >loxP-stop-loxP>LexA: :p65, hs-FLP, FRT40A, lexAop-mCD8: :GFP-insulated spacer-lexAop-rCD2i, and lexAop-rCD2: :RFP-insulated spacer-lexAop-GFPi (*Awasaki et al., 2014*). Transgenes used for Notch depletion include: hs-ATG >KOT > FLP (*Ren et al., 2018*), ase-KD (this study), dpn >FRT-stop-FRT>Cre::PEST (*Awasaki et al., 2014*), actin1loxP-stop-loxPGal4 (this study), UAS-Notch-RNAi (BL#33611), USA-mCD8::GFP (*Lee and Luo, 1999*). vnd-T2A-Gal4, homology arms of about 3 kb each were cloned into pTL1 for knocking-in T2A- Gal4 in vnd with following primers: vnd_55AgeI: TACGACCGGTGATCAAGGAGAACGAGCTATACG; vnd_53StuI: AAGGCCTGGGCCACCAGGCGG; vnd_35PmeI: TACGGTTTAAACTAATATTGCTAGGAACTGGCATTCAC; vnd_33MluI: AAGTACGCGTAACTGGAATAAGTTC. T2A-Gal4 CDS was inserted right after the second last amino acid using traditional Golic heat shock strategy for gene targeting to obtain vnd-T2A-Gal4 transgenic fly (*Rong and Golic, 2000*). ase-KD, the asense promoter (*Jarman et al., 1993*) was put in front of the KD in modified pBPGw through gateway system (Invitrogen) as described previously (*Awasaki et al., 2014*).

actinˈloxP-stop-loxPˈGal4, a synthetic Flox cassette was inserted into a KpnI site between the *actin* promoter and Gal4 as described previously (*Awasaki et al., 2014*).

## MARCM

For ts-MARCM clonal analysis, 0–2 hr old newly hatched larvae with proper genotype were collected and put into vials (80 larvae/vial) containing standard fly food. The larvae were raised at 25℃ until desired stages. Organisms were resynchronized with respect to puparium formation for those clones induced at late larval and early pupal stages. To induce clones, the organisms were heat-shocked at 37℃ for 15–40 min. After heat shock, the organisms were put back to 25℃ until dissection. For Notch depletion, newly hatched larvae with proper genotype were heat shocked at 37℃ for 15 min to induce the activation of lineage restricted driver for clonal labeling and Notch depletion. After heat shock, the larvae were put back to 25℃ until dissection.

## Immunostaining and Confocal imaging

Adult brains were dissected, fixed, and processed as described previously (*Awasaki et al., 2014*). Antibodies used in this study include rabbit anti-GFP (1:1,000, Invitrogen), rat monoclonal anti-mCD8 (1:100, Invitrogen), rabbit anti-RFP (1:1,000, Clontech), mouse monoclonal anti-Bruchpilot, nc82 (1:50, Developmental Studies Hybridoma Bank), Alexa 488, (Invitrogen), Cy3, Cy5 or Alexa 647 (Jackson ImmunoResearch) conjugated anti-mouse, anti-rabbit, and anti-rat antibody (1:500). After immunohistochemistry, brains were post-fixed with 4% PFA in PBS for 4 hr at RT followed by four washes in PBT for 10 mins and then rinsed with PBS. Brain samples were placed on poly-L-lysine-coated cover slips followed by series dehydrated in ethanol baths (30%, 50%, 75%, 95%, and $3 \times 100\%$) for 10 min each and then 100% xylene three times for 5 min each in Coplin jars. Samples were embedded in DPX mounting medium (Electron Microscopy Sciences, Hatfield, PA). Fluorescent signals of whole-mount adult fly brains were acquired at 1 μm intervals using a $40 \times$ C Apochromat water objective (NA = 1.2) and 0.7 zoom factor at $1024 \times 1024$ pixel resolution on Zeiss LSM710 confocal microscope (Carl Zeiss). The 'Janelia Workstation' image-viewing software (Murphy et al., unpublished data) was used to analyze confocal stacks. The whole brain images were registered and aligned to a standard brain (JFRC2010, https://github.com/VirtualFlyBrain/DrosAdultBRAINdomains) using the reference nc82 channel as described previously (*Aso et al., 2014*).

## Neuron type annotation and visualization

Following brain registration, manual annotation of clone morphologies with respect to neuropil structures was carried out using the standard fly brain template harboring predefined neuropil masks. Clones were ascribed to specific lineages based on cell body position and primary neurite trajectory characteristic of each lineage (*Yu et al., 2013*). Clones of the same lineage origin were then clustered into morphological types that target distinct neuropils and/or elaborate differentially within shared targets (*Supplementary file 1*), with each stereotyped pattern seen in at least three samples. We further categorized morphological types of a given hemilineage into few morphological groups based on distinctive group-characteristic features (e.g. common primary neuropil targets).

For cross-comparison and presentation, we segmented out representative neurons via 3D interactive segmentation (*Wan et al., 2012*) and warped segmented neurons into the same standard fly brain through whole-brain alignment (*Rohlfing and Maurer, 2003*). To show single neurons with respect to hemilineage morphology, we created 25 hemilineage masks. The hemilineage masks of the seven lineages composed of dual viable hemilineages were generated by compiling representative A or B neurons of each type together. The lineage masks for the 11 lone hemilineages were derived from merging the first larval-born neuron with its paired NB clone present in isolated twin-spot MARCM clones. 3D rendering for 2D presentation was carried out using VVDviewer (https://github.com/takashi310/VVD_Viewer/releases) (*Wan et al., 2009*).

## Birth order analysis

Custom algorithms were used to present the birth time of neuron types or morphological groups with heatmaps. The neuron-type heatmaps show manually sorted neuron types with actual single-cell numbers (max = 10) recovered from induction at given time points. The birth-order of neuron types was determined to our best judgement, based on relative time of beginning, ending, or peak

recovery using both single-cell and NB clone data. By contrast, the morphological-group heatmaps show computer-sorted morphological groups that often consist of types made in discrete time windows. To reflect the multi-window production due to heterogeneous compositions, we generated group-level heatmaps by (1) identifying well-separated production windows within each group, (2) normalizing single-cell distribution to one for each production window, and (3) sorting morphological groups based on the sample distribution in the first production window. For detail, please see the custom algorithms in supplementary materials.

## Quantitative analysis of single neuron morphology

For pairwise comparison of Vnd neuron types by NBLAST, we selected 464 segmented neurons that were warped into a standard adult fly brain template. We then generated the all-to-all score matrix using the NBLAST package available at GitHub (https://github.com/jefferislab/NBLAST_on-the-fly). Various analyses of the score matrix and visualization of the results were carried out by Python.

To measure the total length of main neuronal trajectory, we first created single-neuron skeletons (trees) using the ImageJ skeletonize3D macro (https://imagej.net/Skeletonize3D). We then detected neuronal cell bodies based on the highest intensity value in the skeletons. After locating the root, we identified leaves and measured the lengths of terminal branches as well as internal segments using the Analyze Skeleton plugin in ImageJ (https://imagej.net/AnalyzeSkeleton). We then took a serial pruning strategy to derive the main trajectory for each single neuron, by removing terminal branches shorter than 10, then 25, and finally 50 μm. We ultimately combined all remaining segments to calculate the total length of main neuronal trajectory.

## Acknowledgements

We thank Janelia Workstation, Janelia FlyLight, and Janelia Fly Core for technical supports. We thank Jens Goldammer for discussions. We thank Crystal Di Pietro and Kathryn Miller for administrative support. This work was supported by Howard Hughes Medical Institute.

## Additional information

### Funding

| Funder | Author |
|--------|--------|
| Howard Hughes Medical Institute | Tzumin Lee |

The funders had no role in study design, data collection and interpretation, or the decision to submit the work for publication.

### Author contributions

Ying-Jou Lee, Ching-Po Yang, Data curation, Formal analysis, Validation, Investigation, Visualization, Methodology; Rosa L Miyares, Conceptualization, Validation, Visualization, Writing - original draft, Writing - review and editing; Yu-Fen Huang, Yisheng He, Qingzhong Ren, Hui-Min Chen, Resources, Methodology; Takashi Kawase, Software, Visualization; Masayoshi Ito, Yoshi Aso, Resources, Visualization; Hideo Otsuna, Resources, Software; Ken Sugino, Conceptualization, Resources; Kei Ito, Conceptualization, Visualization, Writing - review and editing; Tzumin Lee, Conceptualization, Resources, Data curation, Software, Formal analysis, Supervision, Funding acquisition, Validation, Investigation, Visualization, Methodology, Writing - original draft, Project administration, Writing - review and editing

### Author ORCIDs

Qingzhong Ren (ID) http://orcid.org/0000-0001-9633-1477
Ken Sugino (ID) http://orcid.org/0000-0002-5795-0635
Yoshi Aso (ID) http://orcid.org/0000-0002-2939-1688
Tzumin Lee (ID) https://orcid.org/0000-0003-0569-0111

Decision letter and Author response
Decision letter https://doi.org/10.7554/eLife.53518.sa1
Author response https://doi.org/10.7554/eLife.53518.sa2

## Additional files

### Supplementary files

• Supplementary file 1. Lists of annotated neuron types/groups and their sample numbers recovered from serial 2-hr windows for each of the 25 Vnd hemilineages.

• Supplementary file 2. Lists of annotated neuron types and their neuropil innervation patterns for each of the 25 Vnd hemilineages.

• Transparent reporting form

### Data availability

All data generated or analyzed during this study are included in the manuscript and supporting files. Representative segmented neurons for all presented neuron types are available at Virtual Fly Brain (http://virtualflybrain.org/data/Lee_Lineage2020).

The following dataset was generated:

| Author(s) | Year | Dataset title | Dataset URL | Database and Identifier |
|---|---|---|---|---|
| Lee Y-J, Yang C-P, Miyares RL, Huang Y-F, He Y, Ren Q, Chen H-M, Kawase T, Ito M, Otsuna H, Sugino K, Aso Y, Ito K, Lee T | 2020 | 25 Vnd hemilineages | http://virtualflybrain.org/data/Lee_Lineage2020 | Virtual Fly Brain, Lee_Lineage2020 |

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
