## [Decision Letter]

**Acceptance summary:**

Your study of multiple neuroblast lineages in the *Drosophila* adult brain is a tour de force. It allowed you to confirm some of the basic principles that you and others have established for the generation of neurons, such as the temporal specification of neurons as well as the role of Notch in hemilineages. We also appreciated the novel concepts and in particular the evidence for duplicated lineages as well as the repeated nature of some of the series of neurons produced. This work will be undoubtedly a useful resource for many researchers.

**Decision letter after peer review:**

Thank you for submitting your article "Conservation and divergence of related neuronal lineages in the *Drosophila* central brain" for consideration by *eLife*. Your article has been reviewed by two peer reviewers, and the evaluation has been overseen by Claude Desplan as the Reviewing Editor and K VijayRaghavan as the Senior Editor. The reviewers have opted to remain anonymous.

The reviewers have discussed the reviews with one another and the Reviewing Editor, who has added some more comments at the end, has drafted this decision to help you prepare a revised submission.

All reviewers and the editor in particular were extremely impressed with the extent of the work, his thoroughness and the potentially immense source of information that might emerge from these data. The visualization of the data is also extremely impressive.

However, as pointed out by reviewer #1, it is difficult to justify publication of this huge set of data to a broad audience without doing a much better job extracting important rules that might apply to other systems.

– One of these rules is the role of Notch in hemilineages, although of course this is well-established, and your lab has contributed extensively to this concept.

– Reviewer # 1 points out that one “novel finding pertains to that hemilineages with greater axon/dendrite coverage associate with the N^off^ state, implicating N^on^ in inhibiting long-distance targeting”.

If you could repeat at least one of the lineage studies in an all-N^on^ or all-N^off^ context to provide functional data to one of the main observations, this would greatly help the paper to be accepted.

– The other is the sequential production of neurons and the notion that temporal patterning likely plays a very important role. However, you do not provide a molecular support for this for central brain neuroblasts, and it is quite surprising that nothing is known about the factors that might play a role in Type I neuroblasts. I admit that this is inferred from previous work from Chris Doe's lab (and Claude's lab as well) but it would be important to devise a way to access these potential factors. I am sure that you great creativity could allow you to develop an approach to test for genes expressed at different times in different neurons. I would not ask you to do this now as this would require much time, but you must discuss the issue of temporal patterning of Type I central brain neuroblasts.

– There is another point that I find extremely exciting, which is the repeated nature of lineages portions. This is, to a large extent, buried in the massive data and the paper would greatly gain if this concept were to be significantly emphasized. Of course, one question would be whether there is a repeated sequence of temporal transcription factors. This would require a lot of work as we do not know what these tTFs are, but this should be discussed in more details.

– Reviewer #2 wants you to release this data with publication of the work (and I am sure that you intend to do so as you always did in the past) but you have to explain how this will be achieved technically and how it will be matched to the Virtual Fly Brain or other means to allow comparison with other works. This reviewer also rightfully asks for more rigor (and quantification whenever possible) in describing the new neurons.

In summary, this could be in our view very important work if you could help the reader extract important concepts, e.g. the repeated nature of the lineages and at least provide one set of functional data with Notch, which should be feasible in your hands. Otherwise, it would remain a resource aimed at a very small community of scientists interested in fly neural lineages.

I have added a few more technical comments at the end, but reviewer #2 has a long list of very useful comments that must be addressed.

Reviewer #1:

This is important work, and very impressive in its comprehensiveness. I have no major critique of the data presented. However, the study exclusively entails the mapping of lineages, without any functional assessment of genes potentially controlling lineage progression and/or axon/dendrite projections. Moreover, as far as I can judge, they describe no novel technologies.

The only novel finding pertains to that hemilineages with greater axon/dendrite coverage associated with the N^off^ state, implicating N^on^ in inhibiting long-distance targeting. However, this idea is not tested functionally.

In summary, I am not sure that these findings would be of broad interest, and hence be a good fit for the broad readership of *eLife*.

Reviewer #2:

Lee et al. performed a detailed and careful study of 18 out of the approximately 100 neuroblasts in the *Drosophila* adult brain. The work is a true tour de force, where they imaged 5,771 brains and manually annotated 20,916 clones, which they aligned to a standard fly brain, finally defining 467 morphological neuronal types, that include some previously undocumented MB PAM neurons. They also describe some principles in the generation of neurons by neuroblasts, including some rules already documented for other neuroblasts, such as the temporal specification of neuron types, and hemilineage specific cell death, as well as, more novel concepts, such as the generation of "duplicated" lineages from two different neuroblasts. This work will be undoubtedly a useful resource for many researchers. However, for it to be really useful, and also for non-experts to be able to appreciate and follow the manuscript, the following two main points need to be addressed before publication:

1) Most of the descriptions are very qualitative and hard to appreciate. I have no doubt that what the authors claim in the text is true and that for them, as experts who have been looking at these neurons for years, is obvious. However, many of these claims were impossible for me to evaluate. I give below a few examples, with suggestions on how to improve them:

– "we see more focused, shorter-range innervation in most A hemilineages as compared to their paired B lineages". Given that they have already reconstructed the clones, can this be quantified, by for example plotting total cable length or total percentage of brain coverage for hemilineages A and B?

– Subsection “Morphological complexity decreases with birth order” can "exuberant" arborizations be described in more quantitative terms and the differences mentioned analysed statistically?

– "AOTUv4A and LALv1A make similar FB neurons in comparable temporal patterns." and "This phenomenon indicates that the two NBs produce two identical series of PAM neurons." I find nearly impossible to verify these two statements. The authors should do two things. First, they should calculate NBLAST scores, and statistically test their differences before making claims like "similar" or "identical". Second, when comparing two neuron types they should show in the figure a graph with dot plots of the all by all NBLAST pairwise comparisons of the two hemilineages to be compared, as well as a representative image of each, side by side, so this difference can be appreciated. Having to jump from main figures to supplementary figures with tens of images and find the ones that look similar is extremely hard.

2) They describe and annotate many new neurons and neuron types, which has the potential to be very useful but only if this data is released to the community in a sensible format. I have the following suggestions:

– There should be a plan in place, which should be explicitly written in the manuscript under the section of "Data availability" for the release of all of the skeletons and clones that have been mapped to a common fly reference brain. Ideally, these should be uploaded post-publication to the "Virtual Fly Brain": https://www.virtualflybrain.org/site/vfb_site/yourPaper.htm

– The nomenclature of the neurons and neurons types is not consistent. For example, in subsection “Neurons of same hemilineage origin vary in topology” to say that for the nomenclature "typically" or "we may add", is not best practice. The data should be released with a consistent nomenclature that is the same for all neuron types, and is objective, things like "main arborisation neuropile" can be subjective, instead things like first or last arborisation neuropile might be more appropriate. It would indeed be best to discuss with VFB what the most appropriate nomenclature might be.

Added comments from Claude Desplan:

I do not have major concerns with the description of the 18 lineages as well as no major concerns with the observations made in the paper. However, given the density of information available in the manuscript, some parts of the Discussion must be extended (Particularly temporal patterning).

In Figure 2: you discuss a more focused and shorter-range innervation for most N^on^ vs N^off^ lineages. While you mention that this applies for both paired lineages and unpaired lineages ("The correlation of Notch A/B fate with the extent of neurite targeting and innervation extends beyond lineage boundary, and also applies to unpaired hemi-lineages"), this is visually not entirely clear based on the figure. While panels D and P are indeed morphologically similar, panels O, Q and R (N^off^, unpaired) resemble more the morphological pattern of N^on^ sister-hemi-lineages A-G (green), shorter range innervation. H-N seem to be more long-range innervation resembling N^off^ in the paired-lineages. Figure 2H-N should be discussed in more detail to clarify these principles. It is for example unclear to what extend we only have to consider dense/disperse or also targeting range and midline crossing.

Figure 7: Figure 7E-F and H-I is this something unique to the LaLv1 lineage or is this observed for the 7 paired lineages? "Notch modulates temporal patterning as evidenced by unilateral switches in producing distinct neurons on the A or B side. There exist windows when only one hemi-lineage is changing types, such that multiple A or B neuron types are paired with a single B or A neuron type (e.g. Figure 7E-F' and H-I'). "

Figure 9: This is an interesting observation for neuronal temporal patterning; One sister lineage produces neuronal types sequentially while the other produces several types recurrently. You need to discuss these ideas in terms of potential molecular mechanisms.

Figure 12: This is a good overview of the general principles discussed in the text. The Discussion brings up some potential mechanisms underlying these 3 principles; Cas/Svp, Imp/Syp/Chinmo. This figure would benefit from addition of these molecules in a model to support the text in the Discussion. Some additional cartoon on how sequential/recurrent patterning is regulated in sister lineages could be helpful here as well.

---

## [Author Response]

– Reviewer # 1 points out that one “novel finding pertains to that hemilineages with greater axon/dendrite coverage associate with the N^off^ state, implicating N^on^ in inhibiting long-distance targeting”.If you could repeat at least one of the lineage studies in an all-N^on^ or all-N^off^ context to provide functional data to one of the main observations, this would greatly help the paper to be accepted.

We wish we could validate the model and reveal the underlying molecular mechanisms. We utilized Notch RNAi in the NB clone to determine the Notch state of the hemilineages in this study. Unfortunately, this clonal-level phenotypic analysis provides minimal mechanistic insight; rather the N^on^ hemilineage morphology is altered – appearing like N^off^ cells. It would require much more sophisticated systems to monitor and manipulate Notch activity in defined precursors (GMCs) and then follow the resulting phenotypes with single-cell resolution. Regrettably, we are unable to deliver such functional studies within the given timeframe.

Nonetheless, we managed to examine the A/B morphological differences further with quantitative analysis per reviewer #2’s advice (see below for details). We now ascribe the A/B morphological distinctions primarily to the presence of multiple topological classes and/or morphological groups of B neurons as opposed to a single dominant group/class of A neurons (see second paragraph of Discussion). Our model has been therefore updated to propose that Notch suppresses neurite defasciculation and thus promotes joint neurite extension and neuropil innervation by A neurons (see third paragraph of Discussion).

– The other is the sequential production of neurons and the notion that temporal patterning likely plays a very important role. However, you do not provide a molecular support for this for central brain neuroblasts, and it is quite surprising that nothing is known about the factors that might play a role in Type I neuroblasts. I admit that this is inferred from previous work from Chris Doe's lab (and Claude's lab as well) but it would be important to devise a way to access these potential factors. I am sure that you great creativity could allow you to develop an approach to test for genes expressed at different times in different neurons. I would not ask you to do this now as this would require much time, but you must discuss the issue of temporal patterning of Type I central brain neuroblasts.

Thank you for reminding us how little we know about the molecular mechanisms of type I lineage temporal fates. After excluding classical tTFs (except cas/svp’s requirement for triggering Imp/Syp gradients) from post-embryonic type I lineages, we have implicated Imp/Syp RNA-binding proteins (recovered through NB RNAseq) in the temporal patterning of central brain neuronal lineages. However, we don’t know how the opposite Imp/Syp protein gradients can define ~30 serial temporal fates in a single type I lineage. Given such fine temporal patterning, we envision involvement of additional intrinsic mechanisms which remain to be discovered.

We are therefore particularly excited to see recurrent production of related neuron types as an emerging common theme in the temporal patterning of Vnd lineages. Notably, multiple characteristic features may repeat non-coordinatively within a given hemilineage. This phenomenon suggests parallel recurring mechanisms that diversify type I lineage temporal fates in a combinatorial manner (see fifth and sixth paragraphs of Discussion).

– There is another point that I find extremely exciting, which is the repeated nature of lineages portions. This is, to a large extent, buried in the massive data and the paper would greatly gain if this concept were to be significantly emphasized. Of course, one question would be whether there is a repeated sequence of temporal transcription factors. This would require a lot of work as we do not know what these tTFs are, but this should be discussed in more details.

We are also highly enthusiastic about the repeated nature of temporal fates as assessed by morphology. To examine this interesting phenomenon in a more systematic manner, we further compared the morphology of serially derived neuron types using NBLAST. Remarkably, cyclic relatedness emerged as a dominant temporal pattern in 17 of the 25 Vnd hemilineages (e.g. CREa2B and SMPp&v1A in Figure 12E).

As to possible molecular mechanisms, Notch signaling is an obvious candidate. We have previously shown that Notch underlies the alternate production of AL and AMMC projection neurons in the lateral AL B hemilineage. However, we doubt that Notch alone can account for multiple recurring features. Along this perspective and per reviewer #2’s comment on our previous summary figure, we provide additional illustration on potential molecular mechanisms including the known players and to-be-identified recurring temporal factors (see Figure 14C).

Reviewer #2:[…]1) Most of the descriptions are very qualitative and hard to appreciate. I have no doubt that what the authors claim in the text is true and that for them, as experts who have been looking at these neurons for years, is obvious. However, many of these claims were impossible for me to evaluate.

Thank you for pointing out the challenges in 2D presentation of complex 3D neuron structures. To alleviate the deficiency, we provide manual annotation on innervation patterns for individual neuron types (see Supplementary file 1 and 2). We also present segmented neurons with hemilineage masks in the standard fly brain template, to place single neurons in a larger context. And when needed, we show neurons from certain or multiple angles to make our points of argument as clear as possible (e.g. Figure 2S-X” and Figure 9I-P). Finally, we took the advice and conducted 3D neuron comparison by NBLAST, which largely validate our manual annotation on neuron similarities (see below).

I give below a few examples, with suggestions on how to improve them:– "we see more focused, shorter-range innervation in most A hemilineages as compared to their paired B lineages". Given that they have already reconstructed the clones, can this be quantified, by for example plotting total cable length or total percentage of brain coverage for hemilineages A and B?

We gladly took this advice and tried to measure the total length of main trajectory (excluding terminal elaborations) as an estimate for the range of coverage. However, neurons can vary greatly in the branching pattern and length of branches. We found it extremely challenging to selectively and cleanly remove terminal arborization using a universal parameter. We have no sophisticated algorithm to custom-shave individual neurons either. After various attempts, we devised a sequential pruning strategy, aiming to restore the main trajectory by removing minor branches stepwise from short to long using 10, 25, and 50 µm as the length limits. We then measured the total length of the pruned neuron skeleton for individual A or B neuron types. We present the quantification of A versus B neuron types for the seven Vnd lineages with dual viable hemilineages in a boxplot (Figure 13—figure supplement 1). Despite the presence of very broad B neurons paired with skinny A neurons (e.g. Figure 13A), the average total length of B neurons is significantly larger than that of A neurons only in the CREa1 and SMPp&v1 lineages. Meanwhile, we assessed the total length distribution and found a significantly greater total length distribution in the B than A hemilineages (p-value of paired T test = 0.036). This is consistent with presence of multiple morphological groups or topological classes in most B hemilineages as opposed to only one dominant group/class of A neurons per lineage (e.g. Figure 13B).

In light of the above quantification, we speculate that Notch may suppress gross diversity in A hemilineages by inhibiting defasciculation of neurite bundles. Conversely, B neurons may acquire a wide range of topological classes and/or morphological groups due to lack of Notch inhibition on neurite defasciculation. For details, please see the second and third paragraphs of Discussion)

– Subsection “Morphological complexity decreases with birth order” can "exuberant" arborizations be described in more quantitative terms and the differences mentioned analysed statistically?

It is qualitatively clear that the beginning larval-born neurons often acquire wildly projecting processes and many ending neurons show greatly reduced morphology. For instance, we can readily see the elaboration of first larval-born neurons way beyond the collective projections of the many more subsequently born neurons in full twin-spot NB clones, as shown in Figure 3. Further, it would require systematic efforts to achieve informative quantitative analysis on intricate diverse neuron morphology across highly heterogeneous lineages (as described above on the quantification of A/B hemilineage differences). Due to limited time/resources, we decided to keep our description of this easily observable phenomenon at the qualitative level.

– "AOTUv4A and LALv1A make similar FB neurons in comparable temporal patterns." and "This phenomenon indicates that the two NBs produce two identical series of PAM neurons." I find nearly impossible to verify these two statements. The authors should do two things. First, they should calculate NBLAST scores, and statistically test their differences before making claims like "similar" or "identical". Second, when comparing two neuron types they should show in the figure a graph with dot plots of the all by all NBLAST pairwise comparisons of the two hemilineages to be compared, as well as a representative image of each, side by side, so this difference can be appreciated. Having to jump from main figures to supplementary figures with tens of images and find the ones that look similar is extremely hard.

We gladly took this advice and conducted an all-to-all NBLAST pairwise comparison on 464 Vnd neuron types. After eliminating intra-hemilineage comparisons from the top 3% pool (NBLAST scores > 0.3), we obtained 16 groups of related neuron types, which include two groups of analogous FB neuron types made by AOTUv4A and LALv1A as well as five groups of corresponding CREa1A and CREa2A neuron types. Another group consists of dot-to-dot AOTUv3B and AOTUv4B neurons. These results are consistent with our identifying them as “similar” or “equivalent” by manual annotation. We present the all-to-all NBLAST analysis with a new Figure 12 in the added final Results section.

However, due to the availability of only one high-quality segmented neuron per neuron type, we were unable to compare “same” neurons from different samples. From our available data, the comparison between AOTUv3B_M04 and AOTUv3B_M05 yielded the highest score of 0.59, higher than comparisons across corresponding CREa1A and CREa2A neuron types with scores around 0.45. Judging from this, it could be tricky to identify “same” neurons based on NBLAST scores.

2) They describe and annotate many new neurons and neuron types, which has the potential to be very useful but only if this data is released to the community in a sensible format.

We will deposit the segmented neurons into Virtual Fly Brain upon acceptance of the paper. We indicate this plan of data sharing under the section of Data Availability.

I have the following suggestions:– There should be a plan in place, which should be explicitly written in the manuscript under the section of "Data availability" for the release of all of the skeletons and clones that have been mapped to a common fly reference brain. Ideally, these should be uploaded post-publication to the "Virtual Fly Brain": https://www.virtualflybrain.org/site/vfb_site/yourPaper.htm

We have received email advice on how to deposit the data via dropbox or through a web link. In preparation of data transfer, we have realigned the neurons using the JRC2018-UNISEX (ISO38um) template, as requested by Virtual Fly Brain.

– The nomenclature of the neurons and neurons types is not consistent. For example, in subsection “Neurons of same hemilineage origin vary in topology” to say that for the nomenclature "typically" or "we may add", is not best practice. The data should be released with a consistent nomenclature that is the same for all neuron types, and is objective, things like "main arborisation neuropile" can be subjective, instead things like first or last arborisation neuropile might be more appropriate. It would indeed be best to discuss with VFB what the most appropriate nomenclature might be.

We understand the importance of having consistent nomenclature applicable to all neurons with intricate morphology. We would also like to keep the nomenclature simple, yet informative, such that one can readily refer to and reasonably picture the referred neuron. There are therefore two parts in our nomenclature. The main (first) part is consistent and scalable, which informs about hemilineage and topology. Nonetheless, we add a second part in parentheses which allows flexibility, to provide extra morphological info or just previously used names. We say “typically” or “we may add”, because others may include different info for their interests.

We acknowledge the importance of having consensus nomenclature for the community.

However, the 2019 fly neuron nomenclature meeting held at Janelia failed to reach consensus.

We would also like to coordinate with the nomenclature of EM-reconstructed fly brain neurons. However, the currently available hemi-brain EM data do not cover contralateral projections, which are critical for many Vnd neurons. We have therefore postponed the ultimate efforts in unifying the nomenclature of fly brain neurons.

Added comments from Claude Desplan:[…]In Figure 2: you discuss a more focused and shorter-range innervation for most N^on^ vs N^off^ lineages. While you mention that this applies for both paired lineages and unpaired lineages ("The correlation of Notch A/B fate with the extent of neurite targeting and innervation extends beyond lineage boundary, and also applies to unpaired hemi-lineages"), this is visually not entirely clear based on the figure. While panels D and P are indeed morphologically similar, panels O, Q and R (N^off^, unpaired) resemble more the morphological pattern of N^on^ sister-hemi-lineages A-G (green), shorter range innervation. H-N seem to be more long-range innervation resembling N^off^ in the paired-lineages. Figure 2H-N should be discussed in more detail to clarify these principles. It is for example unclear to what extend we only have to consider dense/disperse or also targeting range and midline crossing.

Thank you for the feedback and careful evaluation of the data, we can see your points. Meanwhile, we have revisited the proposed model on how Notch might affect A/B neuronal morphogenesis (see second and third paragraphs of Discussion). As stated above and detailed further below under quantitative analysis of single-neuron morphology, we now propose that Notch strengthens neurite-neurite affinity. Higher affinity in A hemilineages could suppress neurite defasciculation (thus reducing the number of morphological groups; e.g. Figure 2B[green], C[green], F[green], G[green], I, K, L, N) and facilitate extension of long neurite fascicles (e.g. Figure 2I, K, L, M, N). By contrast, reduced affinity in B hemilineages could promote gross diversity through serial defasciculation of primary projections. Further, reduced affinity across sister branches could enhance neurite elaboration within targeted neuropils. In support of this model, Notch may directly act as a cell adhesion molecule or indirectly mediate cell adhesion through up-regulating canonical cell adhesion molecules such as integrins (Murata and Hayashi, 2016). However, additional factors (e.g. neuropil-characteristic topographic maps) might modulate the gross manifestation of Notch’s morphogenetic effects.

Figure 7: Figure 7E-F and H-I is this something unique to the LaLv1 lineage or is this observed for the 7 paired lineages? "Notch modulates temporal patterning as evidenced by unilateral switches in producing distinct neurons on the A or B side. There exist windows when only one hemi-lineage is changing types, such that multiple A or B neuron types are paired with a single B or A neuron type (e.g. Figure 7E-F' and H-I'). "

Per our current annotation of morphological neuron types, all seven paired Vnd lineages exhibit some unilateral temporal changes. However, it is possible we could have failed to recognize all morphological types, especially in hemilineages that target neuropils which lack stereotyped substructural landmarks. Nonetheless, we are very confident in the observed unilateral temporal changes across the LALa1 A and B hemilineages, because these paired hemilineages show comparable levels of high diversity of morphological types (with 29 and 31 annotated neuron types, respectively). Moreover, we are confident that we identified all LALa1A neuron types, as they innervate distinct FB layers.

Figure 9: This is an interesting observation for neuronal temporal patterning; One sister lineage produces neuronal types sequentially while the other produces several types recurrently. You need to discuss these ideas in terms of potential molecular mechanisms.

There does exist recurrent relatedness at the sub-structural level in the AOTUv3 B hemilineage, as highlighted with arrows in Figure 9A-H and further validated by NBLAST analysis (Figure 1—figure supplement 2C). Because multiple features may recur at different frequencies, we could not confidently determine if the paired A/B hemilineages show coordinated or independent reoccurrences. We therefore did not to comment on patterns of recurrences across paired sister hemilineages.

Figure 12: This is a good overview of the general principles discussed in the text. The Discussion brings up some potential mechanisms underlying these 3 principles; Cas/Svp, Imp/Syp/Chinmo. This figure would benefit from addition of these molecules in a model to support the text in the Discussion. Some additional cartoon on how sequential/recurrent patterning is regulated in sister lineages could be helpful here as well.

We gladly took this advice. We provide a new schematic to illustrate possible underlying molecular mechanisms, including known players and the proposed recurring temporal factors.